# Fix Before Search: Benchmarking Agentic Visual Query Pre-processing in Multimodal Retrieval-augmented Generation

**Shenglai Zeng** [* 1]  **Jiankun Zhang** [* 2]  **Kai Guo** [1]  **Xinnan Dai** [1]  **Hui Liu** [1]  **Jiliang Tang** [1]  **Yi Chang** [2]

## Abstract

Multimodal Retrieval-Augmented Generation (MRAG) has emerged as a key paradigm for grounding MLLMs with external knowledge. While query pre-processing (e.g., rewriting) is standard in text-based RAG, existing MRAG pipelines predominantly treat visual inputs as static and immutable, implicitly assuming they are noise-free. However, real-world visual queries are often "imperfect"—suffering from geometric distortions, quality degradation, or semantic ambiguity—leading to catastrophic retrieval failures. To address this gap, we propose **V-QPP-Bench**, the first comprehensive benchmark dedicated to Visual Query Pre-processing (V-QPP). We formulate V-QPP as an agentic decision-making task where MLLMs must autonomously diagnose imperfections and deploy perceptual tools to refine queries. Our extensive evaluation across 46,700 imperfect queries and diverse MRAG paradigms reveals three critical insights: **(1) Vulnerability**—visual imperfections severely degrade both retrieval recall and end-to-end MRAG performance; **(2) Restoration Potential & Bottleneck**—while oracle preprocessing recovers near-perfect performance, off-the-shelf MLLMs struggle with tool selection and parameter prediction without specialized training; and **(3) Training Enhancement**—supervised fine-tuning enables compact models to achieve comparable or superior performance to larger proprietary models, demonstrating the benchmark's value for developing robust MRAG systems The code is available at https://github.com/phycholosogy/VQQP_Bench.

---

[*]Equal contribution [1]Michigan State University [2]Jilin University. Correspondence to: Kai Guo < guokai1@msu.edu>, Yi Chang <yichang@jlu.edu.cn>.

*Proceedings of the 43rd International Conference on Machine Learning*, Seoul, South Korea. PMLR 306, 2026. Copyright 2026 by the author(s).

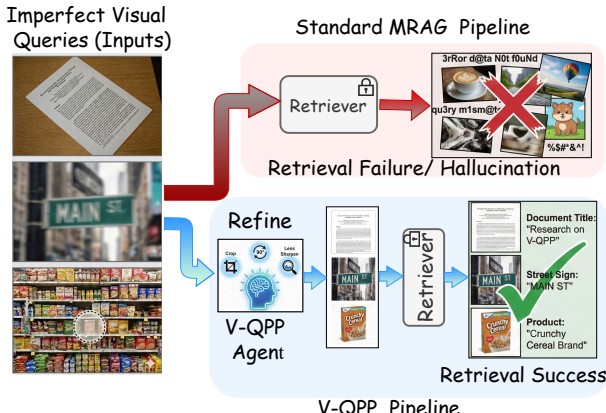

*Figure 1.* **Comparison between Standard MRAG and V-QPP pipelines.** *Top:* When facing real-world imperfections—such as a rotated document, a blurry street sign, or a target product hidden in a cluttered shelf—standard encoders fail to capture the user's intent, resulting in irrelevant retrieval (Red Cross). *Bottom:* The V-QPP pipeline introduces an agentic loop where an MLLM utilizes tools (e.g., cropping the cereal box, deblurring the text) to "clean" the visual query. This active pre-processing step recovers the semantic gap, enabling precise retrieval of the correct entities.

## 1. Introduction

Retrieval-Augmented Generation (RAG) has become the standard approach for reducing hallucinations and providing Large Language Models with external knowledge (Mei et al., 2025). With the rapid evolution of Multimodal Large Language Models (MLLMs) (Alayrac et al., 2022; Li et al., 2023; Team et al., 2023; Yao et al., 2024), this paradigm has naturally expanded into the visual domain, giving rise to Multimodal RAG (MRAG). By empowering models to query external knowledge bases using visual inputs—rather than text alone—Multimodal RAG significantly broadens the scope of AI capability. It serves as a critical bridge between parametric memory and the external knowledge, enabling applications that are difficult to articulate with text, such as analyzing medical images by retrieving relevant clinical cases (Xia et al., 2024), finding visually similar items in e-commerce (Han et al., 2025), or understanding context in visually ambiguous situations(Chang et al., 2022; Wang et al., 2025a). Consequently, the robustness of such systems has become a central research focus.

In a typical MRAG pipeline, the visual queries are inputted

into the retriever to retrieve semantically aligned knowledge (Mei et al., 2025; Hu et al., 2024; Zhang et al., 2025a), implicitly assuming that the input image effectively captures the user's intent. However, in real-world applications, such assumptions do not always hold since visual queries are often "imperfect" due to various real-world factors such as suboptimal capture conditions, environmental clutter, or image degradation (Wang et al., 2025b). For instance, a user photographing a document or a slide often captures it from an oblique angle with heavy perspective distortion; a traveler snapping a landmark from a moving vehicle introduces motion blur; and a user searching for a specific product in a crowded supermarket shelf captures an image dominated by background clutter rather than the target object. Our preliminary study (Section 5.2) shows that these "imperfect" queries will cause the retriever to fetch irrelevant information and leading to generation failures.

In text-based RAG, the necessity of query pre-processing is well-established (Chan et al., 2024; Shi et al., 2026; Xu et al., 2025; Zhang et al., 2025b). Techniques such as query rewriting (Jin et al., 2025; Shi et al., 2026), expansion (Jiang et al., 2025), and decomposition (Xu et al., 2025) are standard components designed to bridge the gap between a user's raw input and the indexed knowledge. In contrast, the "query processing" stage in Multimodal RAG remains surprisingly underexplored. Existing approaches (Yuan et al., 2024; Wu et al., 2025) predominantly treat the visual input as a static, immutable signal, placing the entire burden of robustness on the visual encoder. We argue that this passive approach is insufficient. In text-based RAG systems, LLMs commonly serve as query rewriters to refine and clarify ambiguous user inputs before retrieval (Trivedi et al., 2023; Chan et al., 2024). Analogously, in Multimodal RAG systems, MLLMs should operate as active agents capable of inspecting and refining visual queries by applying appropriate transformations—such as geometric corrections (e.g., re-orienting misaligned pictures) and signal-level adjustments (e.g., cropping to focus on specific regions)—before sending them for retrieval, as shown in Figure 1.

To systematically quantify the impact of imperfect visual queries on MRAG and the potential of visual query pre-processing, we introduce V-QPP-Bench, the first comprehensive benchmark designed to assess Visual Query Pre-processing (V-QPP) in Multimodal RAG systems. We formulate V-QPP as an agentic decision-making task, where models must autonomously diagnose visual imperfections and select appropriate tools to refine the query. Our benchmark comprises 46,700 imperfect visual queries constructed through a rigorous pipeline that combines real-world imperfect images and controlled synthetic augmentations from commonly-used existing datasets (Chen et al., 2023; Wang et al., 2025c; Lerner et al., 2022). It covers a hierarchical taxonomy of 10 imperfection types, ranging from geomet-

ric distortions (e.g., Rotation) and quality degradation (e.g., Noise) to semantic ambiguities (e.g., Multi-object Expansion). Furthermore, the benchmark encompasses 5 diverse MRAG retrieval paradigms, supporting both Text-Targeted (e.g., Image-to-Text) and Vision-Inclusive (e.g., Image-to-Image) retrieval. We integrate these queries into a standardized pipeline to evaluate the agentic capabilities of a wide spectrum of MLLMs, including leading proprietary models (e.g., GPT-4o) and open-source models (e.g., Qwen3-VL).

Our extensive evaluation yields three critical insights:

- **The Vulnerability Gap:** Visual query imperfections severely degrade both retrieval recall and end-to-end MRAG performance, posing significant challenges.
- **The Restoration Potential & Agentic Bottleneck:** Oracle preprocessing can recover near-perfect performance, validating V-QPP's Potential. However, off-the-shelf MLLMs face significant bottlenecks in tool selection and parameter prediction without specialized training.
- **Specialized Training Unlocks V-QPP Capabilities:** Fine-tuning enables compact models to achieve comparable or superior performance to larger proprietary models in visual query preprocessing, demonstrating the benchmark's value for developing robust MRAG systems.

## 2. Related Works

### 2.1. RAG and Query Processing

Previous research has explored various query processing techniques in text-based RAG, which have been proven to be effective in improving retrieval quality and consequently improving RAG performance. Gao et al. (2023) introduced HyDE to bridge the semantic gap in zero-shot retrieval by generating hypothetical documents from queries and using their embeddings for search. Chan et al. (2024) introduced RQ-RAG, which utilizes SFT to train the model to dynamically refine search queries via rewriting, decomposition, and disambiguation. Jiang et al. (2025) proposed DeepRetrieval to optimize query generation via reinforcement learning, demonstrating substantial gains in retrieval recall over previous SOTA and commercial models. In many advanced agentic RAG pipelines(Jin et al., 2025; Shi et al., 2026; Xu et al., 2025; Zhang et al., 2025b), query pre-processing/generation are already a necessity component and be co-optimized in the end-to-end post training. With the rise of MLLMs, RAG has naturally extended to the multimodal domain (MRAG) (Chen et al., 2022; Thiyagarajan, 2025). **However, a critical methodology gap remains:** while text RAG actively refines queries, current MRAG systems predominantly treat visual inputs as static and immutable signals. They implicitly assume canonical visual quality, leaving the system vulnerable to real-world imperfections. Our work introduces V-QPP to address this dispar-

ity, proposing that MLLMs should act as active agents to refine visual queries before retrieval, analogous to rewriting mechanisms in RAG.

## 2.2. Robustness Benchmarks for MLLMs

As MLLMs are increasingly deployed in high-stakes domains, evaluation has shifted from general capabilities (e.g., MME (Fu et al., 2025), MMBench (Liu et al., 2024b)) to safety and robustness. Early works like Adversarial VQA (Li et al., 2021) and NaturalBench (Li et al., 2024) focused on classification robustness against leading questions or logical traps. In the specific context of MRAG, recent benchmarks have begun to address the fragility of retrieval pipelines. **Visual-RAG** (Wu et al., 2025) introduces "hard negatives" to evaluate whether models can resist visual distractors in the retrieval corpus. **CrossCheck-Bench** (Tian et al., 2025) and **ConflictVis** (Liu et al., 2025) systematically diagnose "vision-knowledge conflicts," testing if models hallucinate when parametric knowledge contradicts visual evidence. Furthermore, **CRAG-MM** (Wang et al., 2025b) targets low-quality inputs (e.g., blur, occlusion) in egocentric scenarios, emphasizing the model's ability to perform "negative rejection" (i.e., refusing to answer) when retrieval fails. However, existing benchmarks predominantly adopt passive strategies that either expect encoders to tolerate imperfections or simply reject imperfect queries. In contrast, **V-QPP-Bench** pioneers the evaluation of *active refinement*. We posit that addressing these visual imperfections is crucial for the reliability of real-world MRAG systems; V-QPP-Bench assesses whether an agentic MLLM can autonomously *diagnose* these flaws (e.g., rotation, clutter) and *repair* them via tool use to recover retrieval performance, shifting the paradigm to active perception.

## 3. Problem Formulation

In this section, we formally define the Multimodal RAG retrieval process, model the distribution shift caused by real-world visual imperfections, and formulate Visual Query Pre-Processing (V-QPP) as an agentic decision-making task.

### 3.1. Preliminaries: Unified MRAG Retrieval Paradigms

Let $\mathcal{K} = \{(k_i, t_i)\}_{i=1}^N$ denote a multimodal knowledge base of size $N$, where each entry consists of an **index key** $k_i$ and its corresponding **textual knowledge** $t_i$ (e.g., Wikipedia corpus). The index key $k_i$ is multimodal-flexible, which can be an image, raw text, or image-text pairs (e.g., an image and its associated captions). Given a query $q$, which may be an image $I$ or a composed image-text pair $I \oplus T$, the objective is to identify the most relevant index keys and retrieve the corresponding evidence. This retrieval process

can be uniformly formulated as:

$$\mathcal{D}^* = \{t_i \mid i \in \hat{\mathcal{I}}\}, \quad \hat{\mathcal{I}} = \text{top-}k \left\{ \mathcal{S}(\psi(q), \phi(k_i)) \right\}_{i=1}^N \tag{1}$$

Here, $\phi(\cdot)$ is the **Key Indexing Function** that encodes $k_i$ into a compatible representation, $\mathcal{S}(\cdot, \cdot)$ is the similarity metric, and $\psi(\cdot)$ is the **Query Transformation Function** that maps query $q$ into the retrieval embedding space. The retrieved texts $\mathcal{D}^*$ are then concatenated with $q$ and fed to the MLLM for generation. Our V-QPP benchmark supports five commonly used MRAG retrieval settings, categorized by the *modality of* $k_i$:

- **Category I: Text-Targeted Retrieval.** The index key $k_i$ is text (typically identical to $t_i$). Visual queries are transformed into textual representations for matching against a text corpus:
  (1) *I2T via Dense Encoding:* $\psi$ uses a dual-encoder (e.g., CLIP (Radford et al., 2021)) to map image $I$ to a shared embedding space, where $\mathcal{S}$ computes cosine similarity against $\phi(k)$.
  (2) *Caption-then-Retrieve:* A captioner (e.g., BLIP2 (Li et al., 2023)) first converts $I$ into caption $C$, followed by text-to-text retrieval to find $\mathcal{D}^*$.
- **Category II: Vision-Inclusive Retrieval.** The key $k_i$ is an image or image-text pair. Matching is performed in multimodal spaces with direct visual feature comparison:
  (3) *Image-to-Image (I2I):* $\psi$ encodes the query image into a visual embedding space, where $\mathcal{S}$ retrieves top-$k$ similar images with their associated texts $\mathcal{D}^*$.
  (4) *Composed Retrieval (I+T → I+T):* $\psi$ jointly encodes image and text to query a multimodal database of image-text pairs, returning $\mathcal{D}^*$.
- **Category III: Search API Integration.** Direct use of commercial visual search APIs (e.g., Google Lens):
  (5) *Visual Search via API:* $\psi$ performs web-scale nearest-neighbor search to retrieve visually similar entities $\mathcal{E} = \{e_1, \ldots, e_k\}$, extracting their metadata and descriptions as $\mathcal{D}^*$.

Despite their structural differences, all settings rely on the assumption that the input $I$ effectively captures the user's information need and matches relevant contexts. However, our study reveals that visual imperfections in $I$ significantly degrade retrieval performance across all paradigms.

### 3.2. Modeling Visual Imperfections

In real-world scenarios, users rarely provide canonical queries. We model the visual component of a user query as a corrupted version of the canonical intent, generated by a composition of degradation functions:

$$I_{query} = f_L \circ \cdots \circ f_2 \circ f_1(I_{gt}; \phi_1, \ldots, \phi_L) \tag{2}$$

where each $f_i$ denotes a corruption operation (e.g., rotation, noise, semantic ambiguity) parameterized by $\phi_i$ (e.g.,

rotation angle, noise variance). These imperfections introduce a fundamental distribution shift in retrieval. The transformed representation $\psi(I_{query})$ becomes misaligned with the knowledge index, leading to a significant drop in matching scores across all paradigms:

$$\mathcal{S}(\psi(I_{query}), \phi(k_{gt})) \ll \mathcal{S}(\psi(I_{gt}), \phi(k_{gt})) \qquad (3)$$

Consequently, $k_{gt}$ (the key associated with the ground-truth evidence) often falls outside the top-$k$ retrieved set. For instance, in Category I.1 and II, this manifests as embedding drift; in Category I.2, as hallucinated captions; and in Category III, as failed object detection.

### 3.3. The V-QPP Task

We define Visual Query Pre-Processing (V-QPP) as an agentic task. Instead of passively feeding $I_{query}$ into the transformation function $\psi$, the system employs an MLLM agent $\pi$ equipped with a perceptual toolbox $\mathcal{T} = \{\mathcal{T}_1, \ldots, \mathcal{T}_M\} \cup \{\mathcal{T}_\emptyset\}$, which includes refinement operations (e.g., `rotate`, `crop`, `deblur`) and a "Pass" action $\mathcal{T}_\emptyset$.

Given an imperfect query $I_{query}$, the agent iteratively diagnoses visual imperfections and predicts a sequence of action tuples $\tilde{\mathcal{T}} = [(\mathcal{T}_1, \rho_1), \ldots, (\mathcal{T}_L, \rho_L)]$, where each $\mathcal{T}_j \in \mathcal{T}$ is a selected tool and $\rho_j$ represents execution parameters (e.g., rotation angle, crop coordinates). The refined query is obtained through sequential tool applications: $I' = (\mathcal{T}_L \circ \cdots \circ \mathcal{T}_1)(I_{query}; \{\rho_i\}_{i=1}^L)$.

The objective of the V-QPP agent is to maximize retrieval recall within the MRAG paradigm:

$$\max_{\tilde{\mathcal{T}}} \mathbb{E}_{I_{query}} \left[ \mathbb{I}\left( k_{gt} \in \text{top-}k \left\{ \mathcal{S}(\psi(I'), \phi(k_i)) \right\}_{i=1}^N \right) \right] \qquad (4)$$

This formulation shifts the burden of robustness from the frozen retriever to the pre-processing stage. In this work, the oracle action is restricted to image-space, pixel-level query preprocessing before retrieval, directly analogous to word-level query rewriting in text-based RAG. Feature-level interventions would require modifying the retriever or its embedding space, and are therefore beyond the scope of visual query preprocessing.

## 4. V-QPP-Bench

### 4.1. Data Construction Pipeline

To systematically quantify the impact of imperfect visual queries on MRAG and the V-QPP effect, we construct **V-QPP-Bench**. Unlike traditional robustness benchmarks that only provide noisy images for classification, V-QPP-Bench provides **Refinement Triplets**: $\langle I_{query}, \mathcal{T}^*, I_{gt} \rangle$, consisting of the imperfect query $I_{query}$, the optimal tool execution trace $\mathcal{T}^*$, and the canonical ground truth image $I_{gt}$.

**Data Source.** We select two established knowledge-based VQA benchmarks corresponding to the two MRAG categories: InfoSeek (Chen et al., 2023; Wang et al., 2025c), which provides curated text contexts ideal for Text-Targeted Retrieval, and ViQuAE (Lerner et al., 2022), which offers comprehensive image coverage for Vision-Inclusive Retrieval. Both datasets provide human-annotated question-answer pairs requiring external knowledge retrieval from Wikipedia corpora, with rigorously validated golden context passages containing ground-truth answers. For **Category I** and **Category III**, we adopt the InfoSeek subset from the VRAG task in Wang et al. (2025c), comprising 1,128 original queries. For **Category II**, we utilize 3,542 examples from ViQuAE (Lerner et al., 2022). We denote these original images as original queries $I_{gt}$ and construct imperfect visual queries through systematic perturbations of $I_{gt}$.

**Imperfect Query Generation.** Our data construction follows a "reverse-engineering" paradigm to ensure precise ground truth. We model visual imperfections as transformation functions $f_i(\cdot)$[1]. For a clean image $I_{gt}$, we generate the imperfect query $I_{query} = f_i(I_{gt}; \phi_i)$, where $f_i$ denotes the corruption operations and $\phi_i$ represents the corresponding parameters (e.g., rotation angle $\theta$, crop region $b$). As shown in Table 1, we define a variety of corruption operations to transform source images into imperfect queries, categorized into three types: **(1) Geometric Distortions** include *Rotation*, which randomly rotates images by pre-set angles ($\theta \in \{90°, 180°, 270°\}$), and *Flip*, which applies horizontal or vertical reflections to alter spatial orientation; **(2) Quality Degradation** encompasses *Brightness*, which scales pixel intensity by factor $\beta$ to simulate over/under-exposure, *Blur*, which applies Gaussian kernels with varying kernel sizes to simulate defocus or motion blur, *Noise*, which injects Gaussian noise at different levels to simulate sensor imperfections, and *Crop*, which extracts central regions at various scales to simulate incomplete capture or zooming; and **(3) Semantic Ambiguity** introduces visual clutter through *Expand*, which arranges the target alongside similar objects in a $2 \times 2$ grid layout, *Overlay*, which superimposes distractor objects at specific locations onto the query, *Watermark*, which adds semi-transparent text at the bottom-right corner, and *RealWorld*, which embeds the target within naturally occurring multi-object scenes. The added content is irrelevant to the original query intent by construction: $I_{gt}$ defines the target entity and answerable knowledge, while the added objects, overlays, watermarks, or surrounding scenes are introduced only as distractors. Hence, the oracle refinement removes or suppresses these irrelevant regions, or crops the image back to the target.

To ensure comprehensive evaluation across all imperfec-

---

[1]Although V-QPP-Bench supports compositional corruptions, we focus on single-step cases for fine-grained diagnosis and report multi-step results in Appendix D.3.

*Table 1.* Visual corruption operations for imperfection injection.

| Operation($f$) | Parameters($\phi$) | Description | $f^{-1}$ Tools |
|---|---|---|---|
| Original | None | No transformation | – |
| Rotation | $\theta \in \{90°, 180°, 270°\}$ | Rotate by $\theta$ | $\mathcal{T}_{\text{rotate}}$ |
| Flip | $\tau \in \{\text{h, v, both}\}$ | Axis-aligned flip | $\mathcal{T}_{\text{flip}}$ |
| Brightness | $\beta \in \{0.25, 0.5, 1.5, 1.75\}$ | Scale brightness by $\beta$ | $\mathcal{T}_{\text{lum}}$ |
| Blur | $\sigma \in \{9, 15, 21, 27\}$ | Gaussian blur (kernel $\sigma$) | $\mathcal{T}_{\text{deblur}}$ |
| Noise | $\sigma_n \in \{0.05, 0.1, 0.15, 0.2\}$ | Gaussian noise ($\sigma_n$) | $\mathcal{T}_{\text{denoise}}$ |
| Crop | $s_c \in \{0.4, 0.5, 0.6, 0.7\}$ | Crop $s_c$-proportional region | – |
| Expand | quad $\in \{\text{TL, TR, BL, BR}\}$ | $2 \times 2$ tiling | $\mathcal{T}_{\text{crop}}$ |
| Overlay | $l \in \{\text{TL, TR, BL, BR, C}\}$ $f \in \{0.125, 0.25, 0.5\}$ | Overlay at $l$ (scale $f$) | $\mathcal{T}_{\text{loc}}, \mathcal{T}_{\text{fill}}$ |
| Watermark | font $\in \{1.0, 2.0, 3.0\}$ | Text at bottom-right | $\mathcal{T}_{\text{loc}}, \mathcal{T}_{\text{fill}}$ |
| RealWorld | template $\in \{1, 2, 3, 4\}$ | Embed in screen scene | $\mathcal{T}_{\text{loc}}, \mathcal{T}_{\text{crop}}$ |

tion types, *every* source image undergoes all 10 corruption operations. Consequently, the reverse (oracle) operations achieved by perception tools to maximally recover the imperfect operation, denoted by $f_i^{-1}$ are naturally the optimal refinement action $\mathcal{T}^*$. This allows us to evaluate agent performance not just on final retrieval success, but also the tool selections and parameter predictions. **More detailed examples and discussion about the imperfect operations and reverse (oracle) operations are shown in Appendix A.1**

*Table 2.* Specification of atomic perceptual tools in library $\mathcal{T}$.

| Tool | Parameters | Description |
|---|---|---|
| $\mathcal{T}_{\text{rot}}$ | {"degrees": int} | Rotate clockwise |
| $\mathcal{T}_{\text{flip}}$ | {"direction": str} | Flip (h/v/both) |
| $\mathcal{T}_{\text{lum}}$ | {"factor": float} | Brightness scaling |
| $\mathcal{T}_{\text{deblur}}$ | None | Remove blur |
| $\mathcal{T}_{\text{denoise}}$ | None | Remove Gaussian noise |
| $\mathcal{T}_{\text{loc}}$ | {"prompt": str} | Detect object, return bbox |
| $\mathcal{T}_{\text{crop}}$ | {"bbox": dict} | Crop to bbox region |
| $\mathcal{T}_{\text{fill}}$ | {"bbox": dict} | Fill bbox with white |

**Retrieval Corpus Construction.** For the InfoSeek subset, we employ the curated knowledge base from (Wang et al., 2025c), which provides 904–1,221 context passages (100 words each) per query. This includes: (a) *golden context* containing the answer, and (b) *distractor context* retrieved from Wikipedia using entity keywords from the image. Following (Wang et al., 2025c), we use a filtered subset (not full Wikipedia) to maintain computational tractability while preserving semantic challenge. For ViQuAE (Lerner et al., 2022), we construct a multimodal corpus from Wikipedia image-text pairs ($|\mathcal{K}| = 1,474,173$), excluding cases where query images match database images exactly. To eliminate length-induced retrieval bias, all context passages are strictly normalized to exactly 100 words. Each golden context is further guaranteed to contain both the ground-truth entity depicted in the query image and the precise answer to the associated question, ensuring unambiguous answerability upon successful retrieval. During inference, the retriever returns not only the 100-word text snippet but also the corresponding Wikipedia article title, providing agents with complementary metadata to facilitate entity disambiguation and answer grounding.

## 4.2. The Agentic Environment (Tool Library)

We formulate V-QPP as a tool-use decision process where multimodal large language models (MLLMs) dynamically select preprocessing operations from a standardized library $\mathcal{T}$ (Table 2). Our library comprises eight atomic tools organized into three categories: **(1) Geometric Correction** ($\mathcal{T}_{\text{rot}}, \mathcal{T}_{\text{flip}}$) for lossless spatial adjustments through rotations and flips; **(2) Quality Enhancement** ($\mathcal{T}_{\text{lum}}, \mathcal{T}_{\text{deblur}}, \mathcal{T}_{\text{denoise}}$) for perceptual restoration despite irreversible degradation; and **(3) Semantic Refinement** ($\mathcal{T}_{\text{loc}}, \mathcal{T}_{\text{crop}}, \mathcal{T}_{\text{fill}}$) for compositional clutter removal through detection-crop-masking pipelines. During evaluation, these tools and their descriptions are provided to the MLLM, which must analyze the input visual query and select appropriate preprocessing operations to restore the image. All tools accept JSON-formatted parameters, enabling MLLMs to construct multi-step refinement strategies. Critically, successful V-QPP requires both correct tool selection and precise parameter prediction (e.g., rotation degree, bounding box coordinates). Detailed tool specifications and usage prompts are provided in Appendix A.2 and Appendix A.3.

## 4.3. Evaluation Metrics

To provide a granular analysis of our pipeline, we employ a multi-level evaluation framework covering agentic diagnostics, retrieval recall, and final MRAG performance.

**Level 1: Agentic Diagnostics (Tool & Parameter).** This level assesses whether the MLLM acts as a competent "doctor" to diagnose and repair visual flaws.

- **Tool Selection Accuracy (TSA):** The percentage of cases where the model selects all correct tool types (e.g., choosing Rotate for a rotated image rather than Deblur)[2].
- **Parameter Score (PS):** For the correctly selected tools, we measure the deviation between the predicted parameters and the Oracle parameters.
  - PS is omitted for parameter-free tools (deblur, denoise) and locate (evaluated via downstream crop/fill operations).
  - For discrete-parameter tools (rotate, flip), we report **Mean Accuracy**—the fraction of correctly predicted parameter values.
  - For lum, we report the exponential similarity $\exp(-0.5 \cdot \|\phi_{\text{pred}} - \phi_{\text{gt}}\|_2)$ between predicted and ground-truth brightness factors. [3]
  - For Crop, Fill, we report the **Intersection over**

---

[2] Our tool library is designed to be largely orthogonal — each tool addresses a distinct type of visual degradation with minimal functional overlap. The one exception — 180° rotation being equivalent to combined horizontal and vertical flips — is already handled in our calculation.

[3] We use this normalized similarity rather than raw $L_2$ distance because it is bounded in $[0, 1]$, higher-is-better, and consistent with Mean Accuracy and IoU.

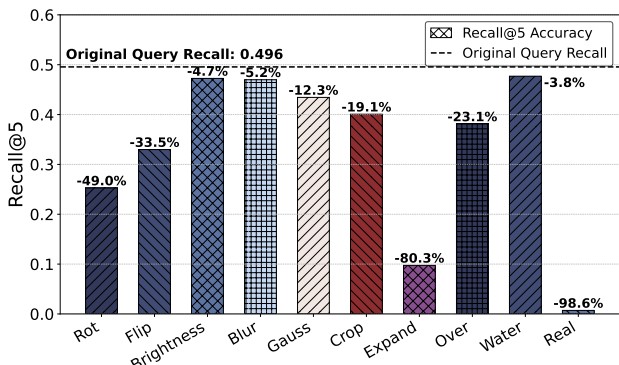

*Figure 2.* Retrieval performance (Recall@5) degradation in MRAG systems under various imperfect query conditions

**Union (IoU)** between the predicted bounding box and the ground truth.

All metrics are designed to be *higher-is-better* with a minimum value of 0. Consequently, when the agent selects a wrong tool, the PS is assigned 0, ensuring that PS reflects both tool selection accuracy and parameter fidelity.

**Level 2: Retrieval Performance.** This metric measures how imperfect queries and pre-processing affect retrieval performance. We evaluate using Recall@K (e.g., R@5), which measures the proportion of queries where the ground-truth document appears in the top-$K$ retrieved results from a database containing golden passages.

**Level 3: End-to-End MRAG Performance.** Finally, we evaluate the final MRAG performance. We majorly report accuracy of **Substring Exact Match** metric, following previous work (Wang et al., 2025c).

## 5. Experiments

In this section, we conduct comprehensive experiments to evaluate the impact of imperfect queries on MRAG and benchmark agentic visual pre-processing abilities. We structure our experiments around three core research questions:

- **RQ1 (Vulnerability):** How do different types of visual imperfections impact the retrieval and final performance of standard MRAG systems?
- **RQ2 (Capability & Gap):** Can current MLLMs effectively diagnose and repair these imperfections to recover performance? What are the promise and bottlenecks of agentic refinement?
- **RQ3 (Learnability):** Can we establish a simple yet effective baseline for V-QPP-Bench? Specifically, does supervised fine-tuning (SFT) enable MLLMs to learn the proposed agentic tasks and improve performance over their zero-shot counterparts?

We begin by introducing the experimental settings in Section 5.1. We then investigate RQ1, RQ2, and RQ3 in Sections 5.2, 5.3, and 5.4, respectively.

### 5.1. Experimental Setup

We primarily report results for the Image-to-Text (I2T) retrieval paradigm; similar patterns across other paradigms are detailed in Appendix D.1. Our evaluation pipeline follows the MRAG workflow: (1) *Visual Query Pre-processing*: the MLLM agent analyzes $I_{query}$ and optionally invokes tools from $\mathcal{T}$ for refinement; (2) *Retrieval*: the refined image $I'$ (or original $I_{query}$) is encoded to retrieve relevant textual knowledge; (3) *Answer Generation*: the MLLM generates answers conditioned on $I'$, question $Q$, and retrieved passages. We evaluate diverse MLLMs:

- **Proprietary models**: GPT-4o (Hurst et al., 2024) and Gemini-2.5-Flash-Lite (Comanici et al., 2025)
- **Open-source models**: Qwen3-VL-4B-Instruct (Yang et al., 2025), DeepSeek-VL2-Small (Wu et al., 2024), Phi-4-Multimodal (Abouelenin et al., 2025), Llama-3.2-11B-Vision (meta, 2024), Mistral-Small-3.1-24B (mistralai, 2025), and Llama3-LLaVA-Next-8B (Liu et al., 2024a)
- **Model scaling**: We evaluate Qwen3-VL variants (4B, 8B, 30B-A3B) (Yang et al., 2025) to examine how model size affects agentic pre-processing capability.

For retrieval, we use `nomic-embed-text-v1.5` (Nussbaum et al., 2024), a multimodal embedding model that projects images and text into a shared 768-dimensional space, with $K = 5$ following standard RAG practice. Ablation studies on alternative retrievers are provided in Appendix D.2. All models receive identical context formatting (Table 5) to ensure fair comparison. We further report a runtime analysis in Appendix D.7, where latency is decomposed according to the three-stage MRAG pipeline: visual query pre-processing, retrieval, and answer generation.

Unless otherwise specified, we adopt forced retrieval as the primary evaluation protocol to isolate the effect of visual query preprocessing. We further report results under an adaptive retrieval setting in Appendix D.6, where the agent decides whether retrieval is necessary.

### 5.2. RQ1: Vulnerability Analysis

In this subsection, we investigate the impact of visual query imperfections on MRAG retrieval performance. Figure 2 shows the average Recall@5 across different types of imperfect queries. The results clearly demonstrate that retrieval performance consistently degrades across all imperfection types, with varying degrees of severity. Most notably, queries with **semantic ambiguities** cause catastrophic performance drops, particularly those containing multiple objects (RealWorld: $-98.6\%$, Expand: $-80.3\%$, Overlay: $-23.1\%$). This suggests that retrievers are easily distracted when images contain multiple competing sub-

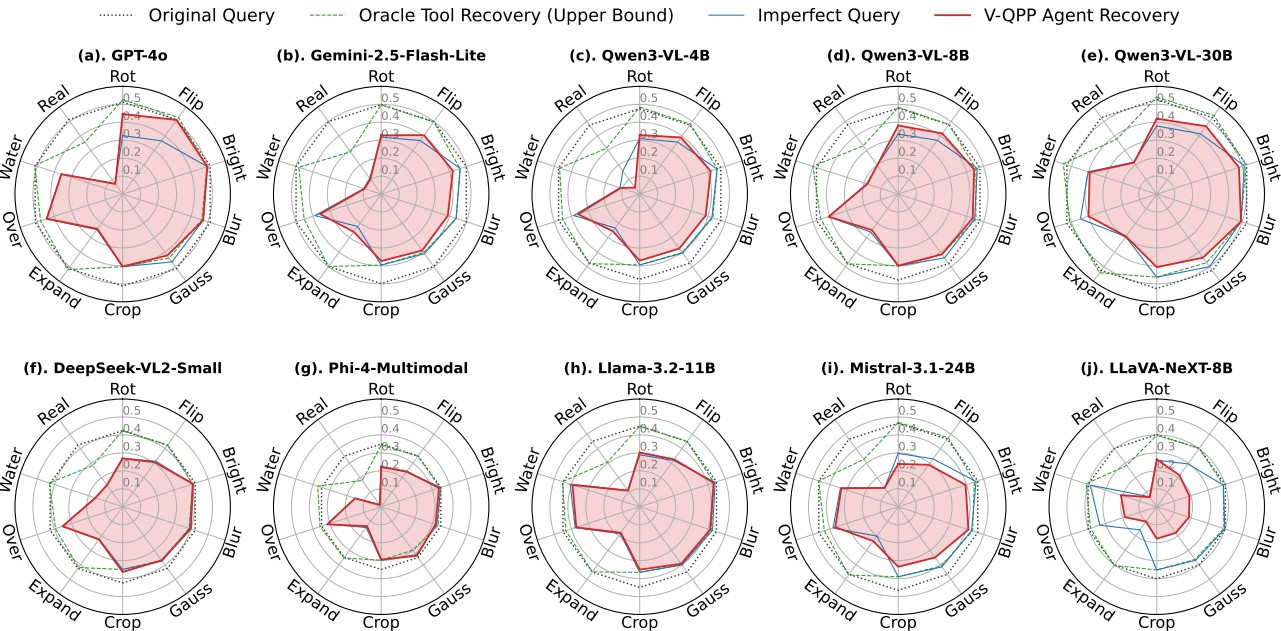

*Figure 3.* **Holistic End-to-End Performance Assessment on V-QPP-Bench.** Visualization of the final MRAG accuracy across various imperfections. **Original Query (Grey Dashed):** Standard MRAG performance on the original queries. **Imperfect Query (Blue Solid):** Standard MRAG performance on the imperfect queries. **V-QPP Agent (Red Solid):** Performance after active agentic refinement. **Oracle (Green Dotted):** Theoretical upper bound using optimal tool transformations.

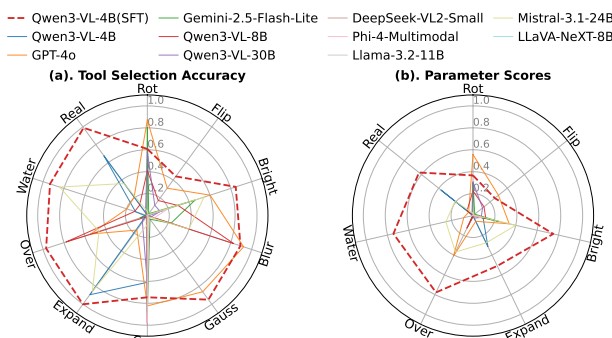

*Figure 4.* Tool selection accuracy and parameter scores across models. The red dot denotes Qwen3-VL-4B-Instruct after SFT, while solid lines represent off-the-shelf models.

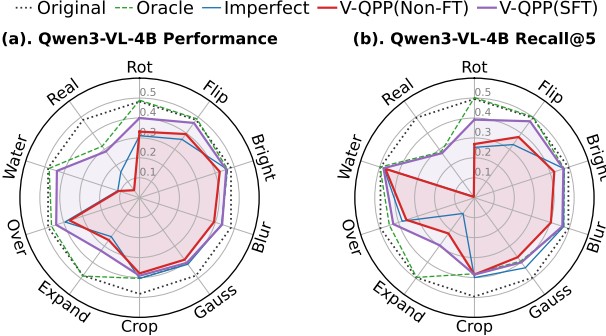

*Figure 5.* Performance and retrieval recall comparison. Purple line: Qwen3-VL-4B-Instruct after SFT; Red line: off-the-shelf Qwen3-VL-4B-Instruct.

jects. In contrast, watermark injection causes minimal retrieval loss, likely because watermarks occupy small regions and retrievers continue to rely on global image features. **Geometric distortions** also significantly impair retrieval, with rotation causing a $49\%$ drop and flip causing a $33.5\%$ drop. Meanwhile, **quality degradations** have relatively modest impacts: brightness ($-4.7\%$), blur ($-5.2\%$), and Gaussian noise ($-12.3\%$). Cropping operations cause moderate degradation ($-19.1\%$), as they introduce substantial information loss and may alter semantic meaning when important features are removed.

Figure 3 presents the final MRAG performance comparison. Compared to original queries (grey), imperfect queries (blue dashed) consistently degrade performance across all

operations and models. Operations causing low recall rates (e.g., RealWorld, Expand, Rotation, Flip) naturally result in severe downstream performance drops. Notably, while watermark insertion minimally impacts recall, it significantly degrades answer quality in some models, likely due to MLLM hallucinations during visual question answering. Conversely, quality degradations such as Gaussian noise, blur, and brightness adjustments cause relatively minor performance reductions. These findings underscore the critical need for robust visual query preprocessing in practical MRAG deployments.

**Observation 1:** Visual imperfections severely degrade both retrieval recall and end-to-end MRAG performance. Semantic ambiguities (RealWorld, Expand) and geometric distortions (Rotation, Flip) cause the most severe failures, while quality degradations (Brightness, Blur, Noise) show relatively modest impacts.

### 5.3. RQ2: Agentic Capability and Bottlenecks

To answer RQ2, we conduct a comprehensive evaluation to benchmark MLLMs' ability to diagnose imperfections and select appropriate tools and parameters for restoring imperfect queries on V-QPP-Bench. Figure 3 shows the end-to-end MRAG performance across different models. Beyond the performance of original queries (grey) and imperfect queries (blue dashed), we present two key results: the *agentic refinement* results (red solid) when different MLLMs serve as V-QPP agents, and the *oracle restoration* results (green dotted), which assume the model always selects the correct tools and parameters (as shown in Table 1 and Appendix A.1). The oracle represents a near-ideal restoration scenario, measuring the upper bound potential of agentic refinement.

The oracle results demonstrate great promise of agentic V-QPP refinement. The results show that applying the correct tools and parameters can effectively recover performance in most cases, particularly for imperfections that cause the most severe degradation (e.g., RealWorld, Rotation, Flip). However, we observe limited or negligible improvement for quality degradation operations (Noise, Blur, Brightness, Crop). This occurs for three reasons: (1) some imperfections cause minimal performance degradation initially (e.g., Brightness), leaving little room for improvement; (2) information loss from cropping is irreversible, as no appropriate restoration tool exists; and (3) denoising and deblurring processes themselves inevitably introduce information loss, limiting their effectiveness. We further observe that oracle processing is not strictly monotonic for every individual sample: although it improves average performance, a small fraction of cases can suffer retrieval decline after correct tool invocation, mainly due to artifacts introduced by restoration tools such as denoising, deblurring, or masking. We provide a detailed analysis in Appendix D.8.

**Observation 2.1:** Oracle results validate substantial restoration potential—correct tool usage can effectively recover performance for severe imperfections (RealWorld, Rotation, Flip), though limited improvement occurs for quality degradations due to minimal initial impact or irreversible information loss.

In contrast to the oracle performance, off-the-shelf MLLMs still face significant bottlenecks in agentic refinement. Even powerful closed-source models (GPT-4o, Gemini-2.5-Flash)

and strong open-source models (Qwen3-VL-30B) show limited improvement over imperfect queries, with gains primarily concentrated on Rotation and Flip queries. More concerning, we observe performance degradation on certain imperfection types for some models (e.g., LLaVA-NeXT-8B), likely stemming from incorrect tool selections or parameter predictions. As shown in Figure 4(a), tool selection accuracy is highly imbalanced across models without training, indicating that most models struggle to correctly identify appropriate tools and instead over-rely on a few specific tools for all imperfect queries. The poor parameter prediction scores in Figure 4(b) further highlight the challenge of correct tool usage. This combination of inadequate tool selection and parameter prediction explains the limited performance improvement and underscores the substantial challenges of the agentic V-QPP task.

**Observation 2.2:** Current MLLMs without V-QPP-specific training face significant bottlenecks—even powerful models show limited improvement due to imbalanced tool selection and poor parameter prediction

### 5.4. RQ3: Improvements via SFT

To answer RQ3, we investigate the learnability of MLLMs in adapting to V-QPP tasks. Specifically, we examine the improvements that supervised fine-tuning brings to V-QPP performance and establish a baseline for our benchmark. We randomly select 100 source images (with their 10 imperfection variants, totaling 1,000 imperfect queries) as the training set, with the remainder serving as the test set. We use imperfect queries as inputs and oracle tools as targets to fine-tune Qwen3-VL-4B-Instruct, training the model to learn tool selection and parameter prediction capabilities, resulting in an enhanced V-QPP agent (V-QPP(SFT)). During inference, we use V-QPP(SFT) as the visual query preprocessor while retaining the original model as the generator to evaluate end-to-end performance. More detailed training information is provided in Appendix C.

The tool usage results are shown in Figure 4 and some examples are shown in Appendix B. V-QPP(SFT) demonstrates consistently high and balanced tool selection accuracy across different imperfection types (Figure 4(a)), with overall accuracy comparable to or exceeding proprietary models such as GPT-4o. The advantage is even more pronounced in parameter prediction scores (Figure 4(b)), where V-QPP(SFT) clearly surpasses all models on most imperfection types, with the only exceptions being Rotation and Flip operations compared to GPT-4o[4]. These results demonstrate

---

[4]For the brightness-adjustment tool, we further compute raw $L_2$ distance as a robustness check. The off-the-shelf V-QPP agent obtains $0.620/2.037$ under exponential similarity / $L_2$ distance, while V-QPP(SFT) obtains $0.884/1.740$, confirming that the im-

that supervised fine-tuning significantly enhances models' tool usage capabilities. Consequently, we observe substantial retrieval and end-to-end performance improvements compared to the off-the-shelf baseline. As shown in Figure 5, V-QPP(SFT) effectively improves both retrieval recall and final MRAG performance under nearly all imperfections compared to its off-the-shelf counterpart, with particularly notable gains for imperfections that cause severe degradations (e.g., RealWorld, Rotation, Expand). These results demonstrate the substantial potential of training on our benchmark to improve V-QPP capabilities, while also establishing a simple yet effective baseline that can be further enhanced through more advanced training paradigms. Notably, this improvement is achieved using only 100 source images, corresponding to 1,000 refinement trajectories, suggesting that V-QPP behaviors are learnable even from a relatively small supervision set rather than requiring large-scale task-specific annotation.

Beyond the main single-step evaluation, Appendices D.3, D.4, and D.5 provide additional analyses of multi-step generalization, out-of-distribution robustness, and preliminary reinforcement learning-based training. Together, these results show that V-QPP(SFT), despite being trained only on single-step corruptions, transfers to unseen compositional and real-world settings; meanwhile, preliminary Group Relative Policy Optimization (GRPO; Shao et al., 2024) training yields modest further gains and points to future opportunities for stronger agent optimization.

> **Observation 3:** Task-specific fine-tuning dramatically improves V-QPP performance, enabling a 4B model to match or surpass larger off-the-shelf models in both tool usage and MRAG performance.

## 6. Conclusion

We present V-QPP-Bench, the first comprehensive benchmark for visual query preprocessing in Multimodal RAG systems. Our evaluation reveals that visual imperfections severely compromise MRAG performance, with semantic ambiguities and geometric distortions causing up to 98.6% retrieval recall drops. While oracle experiments validate substantial restoration potential, off-the-shelf MLLMs struggle with effective query refinement due to poor tool selection and parameter prediction. Through supervised fine-tuning on our benchmark, we demonstrate that even a 4B model can match or surpass larger proprietary models in V-QPP capabilities. V-QPP-Bench establishes a rigorous framework for developing robust visual query preprocessing methods, bridging the gap between MRAG research and real-world deployment. We hope this work catalyzes future research in agentic MRAG systems and robust visual understanding.

---

provement is consistent under both metrics.

## Impact Statement

This paper presents work whose goal is to advance the robustness of multimodal retrieval-augmented generation systems. By systematically evaluating visual query imperfections and preprocessing strategies, our benchmark promotes the development of more reliable vision-language AI systems that perform consistently across diverse real-world conditions. The primary societal benefit of this work is improved accessibility and reduced performance disparities for users with varied image quality constraints. While the preprocessing techniques we evaluate could theoretically be misused in adversarial contexts, we note that our focus is on naturally occurring imperfections and our tools are standard image processing operations publicly available in existing libraries. We release V-QPP-Bench publicly to enable transparent evaluation and encourage the research community to develop robust multimodal AI systems that serve diverse user populations equitably.

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

# A. Details of V-QPP Bench

## A.1. Visual Imperfection Taxonomy and Oracle Operations

Our benchmark adopts a reverse-engineering paradigm in which visual imperfections are formally modeled as parameterized transformation functions $f(\cdot; \phi)$. As summarized in Table 1, we define *ten* corruption operations organized into three distinct categories, complemented by an **Original** baseline that applies the identity transformation ($f(I_{gt}) = I_{gt}$) to assess models' capability in recognizing uncorrupted queries. To ensure comprehensive evaluation while avoiding identical transformations within each operation type, we introduce randomized parameters $\phi$ sampled uniformly from predefined ranges. Consequently, *every* source image undergoes all ten corruption variants with stochastically varied parameters, guaranteeing balanced coverage across the imperfection spectrum.

Below we detail each operation's parameterization $\phi$ and its corresponding oracle operation $f^{-1}(\cdot; \hat{\phi})$, which invokes the perceptual tools defined in Table 2 to restore $I_{query}$ toward the clean ground-truth image $I_{gt}$. These same tools are provided to the agent for autonomous invocation during inference. The critical distinction lies in parameter specification: during oracle-operation evaluation, we supply ground-truth parameters $\phi$ to the tools, thereby establishing the theoretical upper bound of restoration performance achievable by perfect tool selection and parameter prediction. Illustrative examples are shown in Figure 6.

**Geometric Distortions.** These operations alter spatial configuration while preserving pixel-level semantics, making them fully reversible given accurate parameter estimation.

- **Rotation** applies discrete rotations parameterized by $\phi = \{\theta \mid \theta \in \{90°, 180°, 270°\}\}$. We deliberately restrict rotations to these cardinal angles to avoid artifacts such as white borders or unintended cropping that arise at arbitrary angles. Empirical validation confirms that even these geometrically simple transformations pose substantial challenges to both retrieval systems and agent reasoning. In the **real world**, users often capture documents or objects from oblique angles (e.g., photographing while standing off-center), resulting in non-axis-aligned images that misalign with canonical orientations expected by retrieval systems. During the **Oracle Operation**, the ground-truth angle $\theta$ is provided to tool $\mathcal{T}_{\text{rotate}}$, which applies a compensating rotation of $360° - \theta$ to restore $I_{query}$ to its canonical orientation $I_{gt}$.

- **Flip** performs reflections parameterized by $\phi \in \{\text{horizontal}, \text{vertical}, \text{both}\}$. In the **real world**, mobile devices may automatically apply horizontal flipping for front-facing cameras (selfie mode) or misinterpret device orientation during capture, producing mirror-reversed images that invert spatial semantics. During the **Oracle Operation**, the ground-truth flip type $\phi$ is provided to tool $\mathcal{T}_{\text{flip}}$, which reapplies the identical reflection operation to recover $I_{gt}$ from $I_{query}$.

**Quality Degradations.** These operations induce irreversible information loss at the pixel level through processes, making faithful restoration inherently challenging and often limited to approximate recovery.

- **Brightness** scales pixel intensities multiplicatively via $I_{\text{query}} = \beta \cdot I_{gt}$ with $\phi = \{\beta \mid \beta \in \{0.25, 0.5, 1.5, 1.75\}\}$. In the **real world**, uncontrolled lighting conditions (e.g., backlighting, indoor dimness, or direct sunlight) frequently cause underexposure or overexposure, degrading visual discriminability for both humans and machines. During the **Oracle Operation**, tool $\mathcal{T}_{\text{lum}}$ is invoked with the compensating factor $1/\beta$ to approximate restoration. Nevertheless, perfect recovery remains unattainable due to irreversible information loss.

- **Blur** convolves $I_{gt}$ with an isotropic Gaussian kernel of size $k \times k$, where $\phi = \{k \mid k \in \{9, 15, 21, 27\}\}$. In the **real world**, motion blur from handheld shooting or camera shake, as well as defocus due to rapid autofocus failure. During the **Oracle Operation**, tool $\mathcal{T}_{\text{deblur}}$ employs the Richardson–Lucy deconvolution algorithm to approximate restoration. Nevertheless, perfect recovery remains fundamentally unattainable.

- **Noise** injects additive Gaussian noise $\mathcal{N}(0, \sigma^2)$ with $\phi = \{\sigma \mid \sigma \in \{0.05, 0.1, 0.15, 0.2\}\}$. In the **real world**, low-light photography amplifies sensor noise (e.g., photon shot noise), while JPEG compression artifacts introduce blocky distortions—both corrupt high-frequency details essential for fine-grained recognition. During the **Oracle Operation**, tool $\mathcal{T}_{\text{denoise}}$ applies non-local means filtering to suppress noise. However, perfect restoration remains inherently impossible because the stochastic corruption irreversibly destroys the original pixel values, leaving only statistical approximations recoverable.

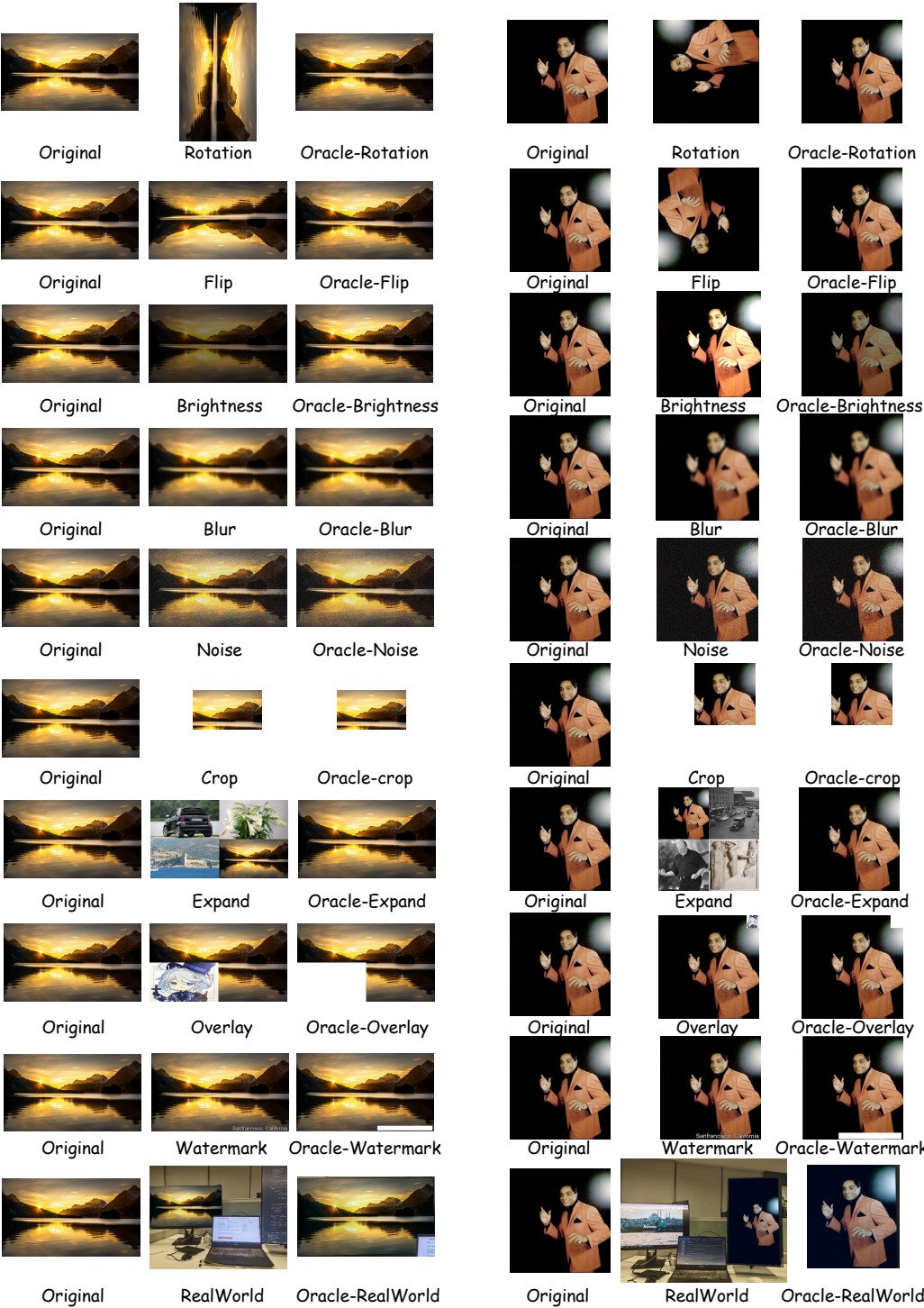

*Figure 6.* Visual illustration of oracle pre-processing across diverse imperfection types and datasets. The left three columns show examples from InfoSeek; the right three columns show examples from ViQuAE. Within each triplet, columns display the ground-truth image $I_{gt}$ (left), the imperfect query $I_{query}$ (middle), and the oracle-restored image $I_{oracle}$ (right) obtained by applying the corresponding tool with ground-truth parameters. Rows from top to bottom demonstrate ten imperfection types: Rotation, Flip, Brightness, Blur, Noise, Crop, Expand, Overlay, Watermark, and RealWorld.

- **Crop** extracts a random region with uniform scaling factor $s$ applied to both width and height ($\phi = s \in \{0.4, 0.5, 0.6, 0.7\}$), discarding peripheral content while preserving aspect ratio. This operation is fundamentally irreversible due to permanent information loss. In the **real world**, users may unintentionally capture only partial objects (e.g., photographing half of a product label) or deliberately crop images for sharing, discarding contextual cues required for accurate retrieval. Consequently, the optimal **Oracle Operation** is a no-op (i.e., abstaining from tool invocation), as any attempt to restore the cropped regions—through inpainting or expansion—would introduce hallucinated content and further degrade fidelity.

**Semantic Ambiguities.** These operations introduce extraneous visual content that obscures the target object, requiring spatial reasoning for disambiguation.

- **Expand** constructs a $2 \times 2$ mosaic by placing $I_{gt}$ alongside three semantically unrelated distractor images, all resized to match $I_{gt}$'s dimensions to ensure uniform grid cells. The target image's position is randomized uniformly across the four grid locations $\phi \in \{\text{TL}, \text{TR}, \text{BL}, \text{BR}\}$ (top-left, top-right, bottom-left, bottom-right). To maintain query-image alignment under this structural transformation, the original query is reformulated as *"According to the [location] part of the image, [original query]"*, where [location] corresponds to the ground-truth grid position of $I_{gt}$. In the **real world**, in cluttered environments (e.g., supermarket shelves or desktop scenes), users capture multiple objects simultaneously but query about only one item, introducing semantic interference from irrelevant co-occurring entities. During the **Oracle Operation**, tool $\mathcal{T}_{\text{crop}}$ performs spatially precise cropping based on $\phi$ to isolate $I_{gt}$ from the composite layout. This operation is fully reversible given accurate positional knowledge.

- **Overlay** superimposes a distractor object onto $I_{gt}$ at location $l \in \{\text{TL}, \text{TR}, \text{BL}, \text{BR}, \text{C}\}$ (top-left, top-right, bottom-left, bottom-right, center) with scale factor $f \in \{0.125, 0.25, 0.5\}$, where both width and height equal $f$ times those of $I_{gt}$ ($\phi = (l, f)$). In the **real world**, foreground occlusions (e.g., fingers covering part of a screen, reflections on glass surfaces, or UI elements overlaid by camera apps) partially obscure target content. During the **Oracle Operation**, tool $\mathcal{T}_{\text{fill}}$ simply masks the overlay region with a solid white patch based on ground-truth parameters $(l, f)$. Although the original content beneath the distractor remains irrecoverable, this minimal intervention effectively eliminates semantic interference by neutralizing the distractor's visual influence, thereby restoring query-image alignment without introducing hallucinated content.

- **Watermark** embeds semi-transparent text "Sanfrancisco, California" at the bottom-right corner with font size parameter $\phi = s \in \{1, 2, 3\}$. In the **real world**, images sourced from the web or institutional repositories frequently contain semi-transparent watermarks, logos, or copyright text that visually compete with primary content during feature encoding. During the **Oracle Operation**, tool $\mathcal{T}_{\text{fill}}$ precisely masks the watermark region with a solid white patch based on the known location and font size. While the underlying pixels cannot be recovered, this minimal intervention effectively neutralizes semantic interference by eliminating the textual distractor, thereby restoring visual focus to the target content.

- **RealWorld** embeds $I_{gt}$ as screen content within synthetically composed cluttered scenes using four predefined templates ($\phi \in \{1, 2, 3, 4\}$), where each template fixes the screen's position and surrounding context. All four templates are visualized in Figure 7. To maintain semantic alignment, queries are reformulated as *"According to the image on [location], [original query]"* with [location] describing the screen's fixed position within the template (e.g., "bottom-left monitor"). In the **real world**, users often photograph content displayed on screens (e.g., slides on a projector, product listings on e-commerce apps), introducing compound distortions including screen glare, moiré patterns, and perspective warping from off-axis viewing. During the **Oracle Operation**, we first invoke tool $\mathcal{T}_{\text{locate}}$—implemented via Grounding DINO (Liu et al., 2023), a phrase-grounding model that localizes visual regions matching textual descriptions—by providing the location phrase corresponding to template to obtain the screen's bounding box coordinates. Subsequently, tool $\mathcal{T}_{\text{crop}}$ crops the region to recover $I_{gt}$.

## A.2. Details about the Tool Library

**Tool Description.** Our perceptual tool library $\mathcal{T}$ comprises eight atomic operations designed to address the three categories of visual imperfections defined in Table 1. Each tool has strictly specified inputs, parameters, and outputs.

- **Geometric Correction Tools:**

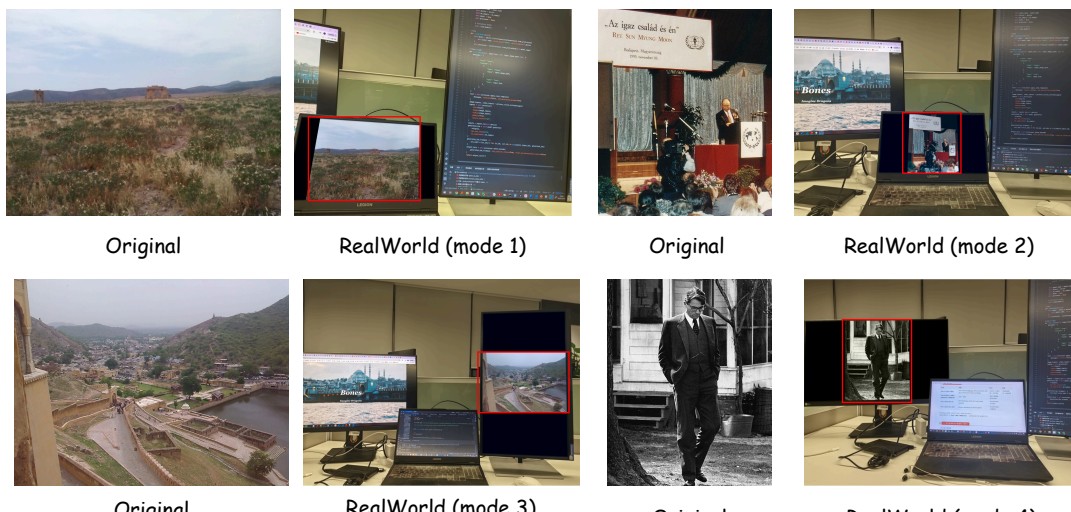

*Figure 7.* Four `RealWorld` template configurations arranged in a $2 \times 2$ grid. Each pair displays the ground-truth image $I_{gt}$ (left) and the corresponding imperfect query $I_{query}$ (right), with the embedded $I_{gt}$ region highlighted by a red bounding box.

- $\mathcal{T}_{\texttt{rotate}}$: Implements lossless rotation based on cv2.rotate. Requires an integer parameter **degrees**, strictly constrained to $\{90, 180, 270\}$. Returns the rotated image.
  - $\mathcal{T}_{\texttt{flip}}$: Performs axis-aligned flipping via cv2.flip. Requires a **direction** parameter accepting `"horizontal"` (left-right flip), `"vertical"` (up-down flip), or `"both"` (combined horizontal and vertical flip). Returns the flipped image.
- **Quality Enhancement Tools:**
  - $\mathcal{T}_{\texttt{lum}}$: Requires a **factor** parameter that multiplicatively scales all pixel values to adjust brightness ($I' = \text{clip}(\textbf{factor} \cdot I, 0, 255)$). Returns the brightness-adjusted image.
  - $\mathcal{T}_{\texttt{deblur}}$: Accepts no parameters. Implements non-blind deblurring based on Richardson-Lucy iterative deconvolution. The core pipeline includes: (1) generating a $15 \times 15$ Gaussian point spread function (PSF); (2) performing 30 iterations of deconvolution; (3) handling boundaries via reflection padding. This implementation demonstrates recovery capability for Gaussian blur and mild motion blur. Returns the deblurred image.
  - $\mathcal{T}_{\texttt{denoise}}$: Accepts no parameters. Employs OpenCV's fast non-local means denoising. The algorithm performs weighted averaging over similar image patches to suppress noise while preserving edge structures. Returns the denoised image.
- **Semantic Refinement Tools:**
  - $\mathcal{T}_{\texttt{locate}}$: Integrates a pre-trained Grounding DINO model (ViT-B/16 backbone, fine-tuned on COCO). Accepts a textual **prompt** parameter specifying the target object to localize. Returns a dictionary containing the top-left and bottom-right coordinates of the highest-confidence bounding box: {`"top_left"`: `[x1, y1]`, `"bottom_right"`: `[x2, y2]`}. This tool enables precise object localization and is typically invoked prior to **crop** or **fill** operations to obtain region coordinates.
  - $\mathcal{T}_{\texttt{crop}}$: Requires a bounding box dictionary with top-left and bottom-right coordinates. Performs precise region extraction, retaining only content within the specified box while completely discarding external regions. Ideal for isolating target objects and eliminating background distractions.
  - $\mathcal{T}_{\texttt{fill}}$: Requires a bounding box dictionary with top-left and bottom-right coordinates. Fills the specified region with solid white (RGB $= [255, 255, 255]$) while preserving the original image dimensions. Suitable for masking watermarks, logos, or irrelevant distractors without introducing spatial distortion from cropping.

Through appropriate composition of these tools, imperfect images can be partially or fully restored to their canonical forms. Appendix A.1 details the oracle operations corresponding to each imperfection type. These tools are also provided to agents during evaluation to assess their capability in autonomously selecting and executing appropriate preprocessing operations for visual query refinement.

---

**Algorithm 1** MRAG Evaluation Pipeline with V-QPP

---

**Require:** Query image $I_{\text{query}}$, question $Q$, tool library $\mathcal{T}$, retrieval backend $\mathcal{R}$, force_retrieval = True
**Ensure:** Final answer $A$, operation log $\mathcal{O}$, retrieval flag $r$
1: $I' \leftarrow I_{\text{query}}$ {Initialize refined image}
2: $\mathcal{O} \leftarrow \text{Agent}_{\text{V-QPP}}(I_{\text{query}}, Q)$ {Stage 1: Tool selection}
3: **if** $\mathcal{O} \neq \emptyset$ **then**
4:     **for** each operation $o \in \mathcal{O}$ **do**
5:         $I' \leftarrow \text{Execute}(o.\text{tool}, I', o.\text{params})$ {Apply tool sequentially}
6:     **end for**
7: **end if**
8: **if** force_retrieval **then**
9:     $r \leftarrow True$ {Stage 2: DEFAULT: Force retrieval}
10: **else**
11:     $r \leftarrow \text{Agent}_{\text{retrieval}}(I', Q)$ {Stage 2: Decide retrieval}
12: **end if**
13: **if** $r$ **then**
14:     $\mathcal{C} \leftarrow \mathcal{R}(I')$ {Retrieve top-$K$ contexts}
15: **else**
16:     $\mathcal{C} \leftarrow \emptyset$
17: **end if**
18: $A \leftarrow \text{Agent}_{\text{gen}}(I', Q, \mathcal{C})$ {Stage 3: Answer generation}
19: **return:** $\{A, \mathcal{O}, r, \mathcal{C}\}$

---

### A.3. Details about the Agentic Environment

**Pipelines** We implement a unified evaluation pipeline that integrates visual query pre-processing (V-QPP), retrieval, and answer generation into a cohesive workflow. The pipeline processes each sample $\langle I_{query}, Q \rangle$ through three sequential stages:

**Stage 1: Visual Query Pre-processing (V-QPP):** The MLLM agent receives the raw query image $I_{query}$ and associated question $Q$. Following the structured prompt in Table 3, the agent analyzes visual imperfections and outputs a JSON-formatted operation sequence $\mathcal{O} = [o_1, o_2, ..., o_k]$, where each $o_i = (\texttt{tool}, \texttt{params})$ specifies a perceptual tool and its execution parameters. The operation sequence is parsed and executed sequentially using the tool implementations defined in Section A.2, yielding the refined query image $I'$. If $\mathcal{O} = \emptyset$, $I'$ defaults to $I_{query}$.

**Stage 2: Retrieval:** To isolate the causal impact of visual pre-processing on retrieval quality, our primary evaluation adopts a *forced retrieval* protocol: the refined image $I'$ is always submitted to the **Retriever** to obtain top-$K$ context passages $\mathcal{C} = \{c_1, ..., c_K\}$, irrespective of whether external knowledge is strictly necessary for answering $Q$. This design eliminates confounding effects from retrieval decisions, enabling a pure assessment of V-QPP's contribution to MRAG robustness. Additionally, to simulate realistic deployment scenarios where unnecessary retrieval may introduce noise, we implement an *agent-decided retrieval* mode. In this variant, the agent evaluates retrieval necessity using the prompt in Table 4, producing a binary decision need_retrieval $\in \{\text{True}, \text{False}\}$ that gates the retrieval operation.

**Stage 3: Answer Generation:** The final answer is generated by an MLLM conditioned on the refined visual input $I'$, the original question $Q$, and (if retrieved) the context $\mathcal{C}$. The generation follows standard VQA protocols with constrained decoding to ensure factual grounding when $\mathcal{C}$ is available.

The complete pipeline execution is formalized in Algorithm 1. Critically, our design isolates the V-QPP module as a *pre-retrieval* operation—any image transformation occurs *before* the query enters the retrieval system. This architectural choice ensures that improvements in retrieval quality can be directly attributed to visual query refinement rather than post-hoc compensation during generation. Furthermore, we log all intermediate artifacts (operation sequences, refined images, retrieval decisions, and contexts) to enable fine-grained analysis of failure modes and success patterns across imperfection types.

Unless otherwise specified, all MLLM inference calls in our evaluation, including V-QPP operation generation and final

answer generation, are served with SGLang using deterministic decoding. For experiments involving agent-decided retrieval, the same decoding configuration is also used for retrieval-decision generation. Specifically, we use greedy decoding with temperature set to 0.0, top-$p$ set to 1.0, and a maximum generation length of 1,024 tokens. All other serving and decoding parameters are kept at the default settings of SGLang (Zheng et al., 2024a). This deterministic setting reduces generation variance and improves reproducibility across models and imperfection types.

In our implementation, coordinates are always interpreted with respect to the current image state. Only CROP and FILL require bounding-box coordinates, and our prompt instructs the agent to call LOCATE before using either tool. Since tool operations are executed sequentially, LOCATE is applied after all preceding transformations have already updated the image. The returned bounding box is therefore aligned with the actual pixel coordinates of the current image, avoiding coordinate-shift issues caused by earlier operations such as rotation, flipping, or cropping. As a result, the main challenge is not coordinate-frame conversion, but whether the agent provides an accurate localization prompt to LOCATE.

## B. Case Studies

This section discusses several cases illustrating why imperfect images fail in retrieval or answer generation, and how SFT-trained models successfully recover performance by invoking appropriate tools. Each example includes the question, original image, imperfect image, retrieved context, and final answer.

For Rotation, as shown in Figure 8, although no pixels are lost, the embedding model may fail to handle orientation changes correctly, causing the embedding representation to shift and resulting in retrieval failure. Applying the rotation tool to restore the original orientation resolves this issue. The Flip operation exhibits similar behavior, as shown in Figure 9.

"question": "What country does this lake belong to?"

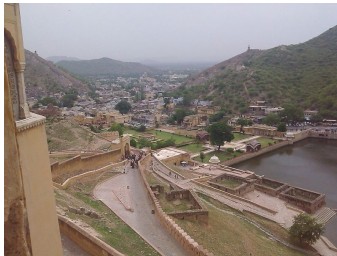 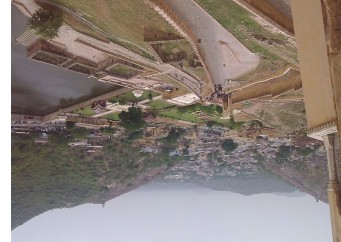 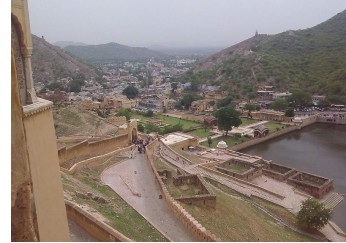

- Image: Original
- Context:
   title: Lake Manasarovar
   text: Maota Lake is located in Amber, in the Indian state of Rajasthan.

- Answer: India

- Imperfect Image: Rotation
- Context:
   title: Gonagolla Vihara
   text: Although it is believed that some constructions in the Vihara belong to the third century BC.
- Answer: Pakistan

- Image: SFT Processed
- Context:
   title: Lake Manasarovar
   text: Maota Lake is located in Amber, in the Indian state of Rajasthan.

- Answer: India

*Figure 8.* Rotation case: A $180°$ rotation preserves all pixels but shifts the embedding distribution, causing retrieval failure. Applying $\mathcal{T}_{\texttt{rot}}$ with correct angle restores the canonical orientation and recovers retrieval performance.

For Brightness, as shown in Figure 10, brightness adjustment causes minor information loss and alters semantic representation, preventing correct retrieval. After brightness restoration, although the image is not fully recovered, the system can still retrieve relevant context successfully.

For Expand, as shown in Figure 11, additional images in the query interfere with the original semantic information. Cropping out the irrelevant regions restores normal retrieval performance.

For RealWorld, as shown in Figure 12, other objects in the scene interfere with the semantics of the target image. Using the locate tool to identify the target region followed by cropping effectively eliminates this interference.

"question": "What country does this lake belong to?"

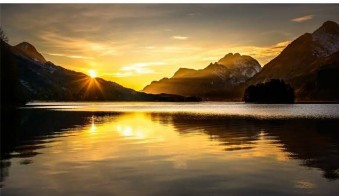 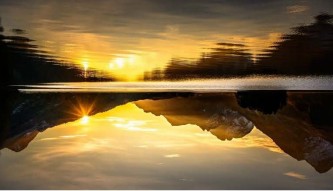 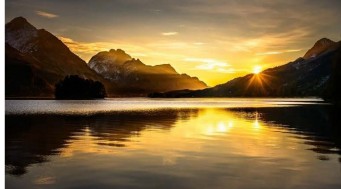

- Image: Original
- Context:
  title: Lake Sils
  text: Lake Sils (German "Silsersee", Romansh: "Lej da Segl") is a lake in the Upper Engadine valley, Grisons, Switzerland.
- Answer: Switzerland

- Imperfect Image: Flip
- Context:
  title: Condit Hydroelectric Project
  text: The White Salmon is a glacier-fed river originating on the slopes of Mount Adams and emptying into the Columbia River..
- Answer: Can not answer

- Image: SFT Processed
- Context:
  title: Lake Sils
  text: Lake Sils (German "Silsersee", Romansh: "Lej da Segl") is a lake in the Upper Engadine valley, Grisons, Switzerland.
- Answer: Switzerland

*Figure 9.* Flip case: Vertical flipping inverts spatial semantics and disrupts feature alignment. The SFT agent correctly applies $\mathcal{T}_{\texttt{flip}}$ with direction "vertical" to recover the original layout and enable accurate retrieval.

"question": "Who designed this building?"

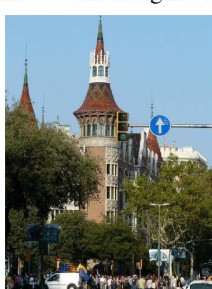 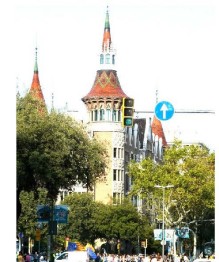

- Image: Original
- Context:
  title: Casa de les Punxes
  text: The Casa de les Punxes or Casa Terradas is a building designed by the Modernista architect Josep Puig i Cadafalch
- Answer: Josep Puig i Cadafalch

- Imperfect Image: Brightness
- Context:
  title: Art Nouveau
  text: Besides the dominating presence of Gaudí, Lluís Domènech i Montaner also used Art Nouveau in Barcelona in buildings …
- Answer: Lluís Domènech i Montaner

- Image: SFT Processed
- Context:
  title: Casa de les Punxes
  text: The Casa de les Punxes or Casa Terradas is a building designed by the Modernista architect Josep Puig i Cadafalch
- Answer: Josep Puig i Cadafalch

*Figure 10.* Brightness case: Severe overexposure alters luminance distribution and degrades discriminative features. Partial restoration via $\mathcal{T}_{\texttt{lum}}$ suffices to shift the embedding back toward the relevant cluster, enabling successful retrieval despite incomplete visual recovery.

"question": "[According to upper left part of the image, ] What country does this building belong to?"

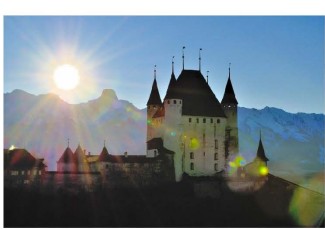
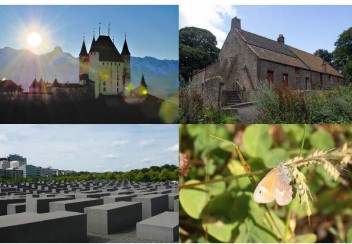
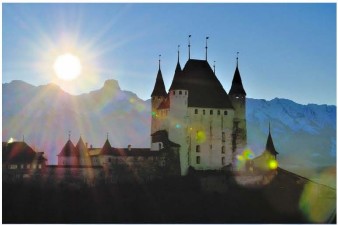

- Image: Original
- Context:
  title: Thun Castle
  text: Thun Castle () is a castle in the city of Thun, in the Swiss canton of Bern.
- Answer: Switzerland

- Imperfect Image: Expand
- Context:
  title: Reinhardsbrunn
  text: Listed in 1891 as one of the artistic landmarks of the duchy, and in 1980 as a landmark of national significance by East Germany
- Answer: Germany

- Image: SFT Processed
- Context:
  title: Thun Castle
  text: Thun Castle () is a castle in the city of Thun, in the Swiss canton of Bern.
- Answer: Switzerland

*Figure 11.* Expand case: A $2 \times 2$ grid introduces semantic interference from distractor objects. Cropping the target region directly eliminates distractions and restores precise retrieval.

"question": " "[According to left screen, ] What country does this lake belong to?"

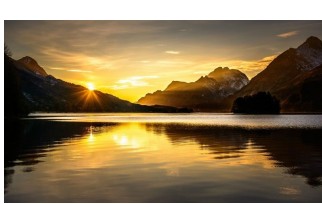
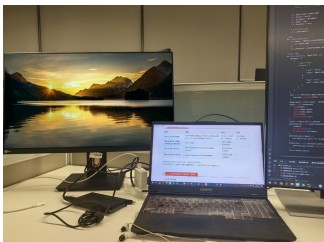
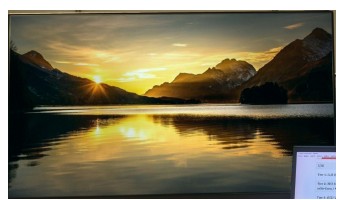

- Image: Original
- Context:
  title: Lake Sils
  text: Lake Sils (German "Silsersee", Romansh: "Lej da Segl") is a lake in the Upper Engadine valley, Grisons, Switzerland.
- Answer: Switzerland

- Imperfect Image: RealWorld
- Context:
  title: CER-10
  text: The first CER-10 system was located at the SKNE (Federal secretary of internal affairs) building in 1961.
- Answer: Can not answer

- Image: SFT Processed
- Context:
  title: Lake Sils
  text: Lake Sils (German "Silsersee", Romansh: "Lej da Segl") is a lake in the Upper Engadine valley, Grisons, Switzerland.
- Answer: Switzerland

*Figure 12.* RealWorld case: The target image embedded in a cluttered scene (e.g., a screen on a desk) suffers from background interference. Localization and cropping ($\mathcal{T}_{\text{loc}} \to \mathcal{T}_{\text{crop}}$) isolate the screen content, removing contextual noise and enabling correct retrieval.

For Overlay, as shown in Figure 13, the overlay not only causes partial information loss but also introduces new semantic interference. By locating the overlay region and applying the fill tool to mask it with pure white, the system can successfully retrieve correct content despite some information loss.

"question": "What country does this lake belong to?"

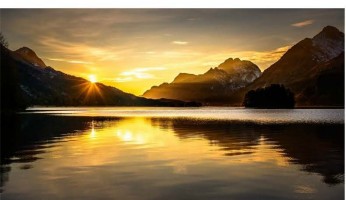 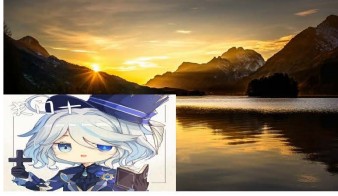 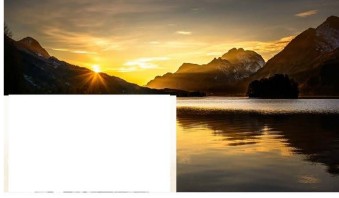

- Image: Original
- Context:
   title: Lake Sils
   text: Lake Sils (German
   "Silsersee", Romansh: "Lej
   da Segl") is a lake in the
   Upper Engadine valley,
   Grisons, Switzerland.
- Answer: Switzerland

- Imperfect Image: Overlay
- Context:
   title: Totemsky District
   text: In particular, Lake
   Sonduzhskoye, by far the
   biggest lake in the district,
   is the source of the Kuloy.
- Answer: Can not answer

- Image: SFT Processed
- Context:
   title: Lake Sils
   text: Lake Sils (German
   "Silsersee", Romansh: "Lej
   da Segl") is a lake in the
   Upper Engadine valley,
   Grisons, Switzerland.
- Answer: Switzerland

*Figure 13.* Overlay case: A distractor image occludes the target and injects spurious semantics. The agent locates the occlusion and applies $\mathcal{T}_{\text{fill}}$ to mask it with white, suppressing interference and allowing retrieval based on intact regions.

For Watermark, as shown in Figure 14, watermarks have minimal impact on semantic information and thus limited effect during retrieval. However, during answer generation, the watermark text "San Francisco, California" misleads the model into incorrectly assuming the image was captured in the United States, resulting in a wrong answer. Removing the watermark via the fill tool enables the model to generate the correct answer.

## C. Experimental Settings for Supervised Fine-Tuning

Motivated by the vulnerability analysis in Section 5.2, which revealed that visual imperfections severely degrade both retrieval recall and end-to-end MRAG performance, we investigate whether models can be trained to actively repair such defects through agentic tool usage. To isolate the contribution of visual pre-processing, we focus fine-tuning exclusively on **Stage 1** (V-QPP)—the tool selection and parameter prediction module—while retaining the original model for retrieval and generation stages. Specifically, we fine-tune Qwen3-VL-4B-Instruct to produce a V-QPP(SFT) agent. During inference, V-QPP(SFT) serves solely as the visual pre-processor: it refines $I_{query}$ into $I'$ via tool execution, while the subsequent retrieval and answer generation stages employ the original, unmodified MRAG pipeline. This design enables a clean attribution of performance gains to the pre-processing module alone.

**Dataset Construction**    Training samples are constructed by formatting the imperfect query image $I_{query}$ and associated question $Q$ according to the structured prompt in Table 3; the target output is the oracle operation sequence $\mathcal{O}^* = [(\texttt{tool}, \hat{\phi})]$ derived from the ground-truth restoration procedures defined in Appendix A.1. We randomly select 100 source images (each with 10 imperfection variants, yielding 1,000 imperfect queries) as the training set, with the remainder reserved for testing. A representative training sample is illustrated in Figure 15.

**Training Parameters.**    We fine-tune Qwen3-VL-4B-Instruct on V-QPP-Bench using supervised fine-tuning with LoRA (Hu et al., 2022) via the LLaMA Factory framework (Zheng et al., 2024b). Key hyperparameters include: LoRA rank $r = 8$ applied to all linear layers (`lora_target=all`), learning rate $1 \times 10^{-4}$ with cosine decay, and 3 training epochs over 1,000 refinement triplets from our benchmark. To accommodate lengthy prompts containing complete tool specifications

"question": "What country does this lake belong to?"

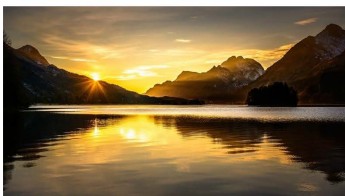
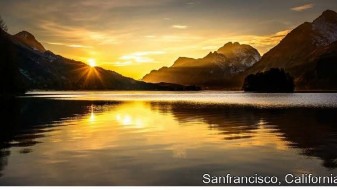
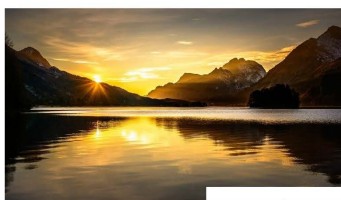

- Image: Original
- Context:
    title: Lake Sils
    text: Lake Sils (German "Silsersee", Romansh: "Lej da Segl") is a lake in the Upper Engadine valley, Grisons, Switzerland.
- Answer: Switzerland

- Imperfect Image: Watermark
- Context:
    title: Lake Sils
    text: Lake Sils (German "Silsersee", Romansh: "Lej da Segl") is a lake in the Upper Engadine valley, Grisons, Switzerland.
- Answer: United States

- Image: SFT Processed
- Context:
    title: Lake Sils
    text: Lake Sils (German "Silsersee", Romansh: "Lej da Segl") is a lake in the Upper Engadine valley, Grisons, Switzerland.
- Answer: Switzerland

*Figure 14.* Watermark case: While retrieval remains largely unaffected, the watermark text "San Francisco, California" misleads the VQA model during answer generation. Removing the watermark via $\mathcal{T}_{\text{fill}}$ eliminates this confounding cue and yields a factually correct answer.

and parameter schemas, we set the maximum sequence length to 65,536 tokens. Training employs an effective batch size of 8 (per-device batch size 1 with gradient accumulation over 8 steps) under bfloat16 precision. The model template (qwen3_vl_nothink) suppresses spontaneous chain-of-thought generation to encourage direct JSON-formatted tool predictions compatible with our execution pipeline.

## D. More Results

### D.1. Other paradigm Results

In Section 5.1, we primarily report results under the Image-to-Text (I2T) dense encoding paradigm. However, as formalized in Section 3.1, MRAG systems operate across five distinct retrieval paradigms categorized by the modality of the matching target. To demonstrate the generality of our findings, we evaluate V-QPP-Bench on the remaining four paradigms below. All experiments follow the same three-stage pipeline (pre-processing → retrieval → answer generation) and employ identical tool libraries and evaluation protocols as the main experiments.

**Caption-then-Retrieve (I2T via Captioning).** This paradigm first converts the visual query into a textual description using a captioning model, then performs text-to-text retrieval. Specifically, we employ BLIP-2 (Li et al., 2023) with the prompt "Caption the image" to generate a descriptive caption $C$ for each query image. The caption is subsequently used to retrieve relevant passages from the text corpus via dense embeddings produced by the Nomic model. Following the main experiments, we adopt the InfoSeek dataset (Chen et al., 2023) with identical train/test splits. Results in the top row of Figure 17 reveal severe vulnerability to visual imperfections: imperfect queries cause substantial recall degradation across all defect types, with RealWorld queries suffering near-total collapse (Recall@5 ≈ 0%) due to captioners hallucinating irrelevant content from cluttered screen-captured scenes. Zero-shot V-QPP agents fail to mitigate this degradation, often selecting inappropriate tools that further distort semantic content. In stark contrast, the SFT-trained agent (V-QPP(SFT)) dramatically recovers performance—particularly for RealWorld queries, where Recall@5 rises from near-zero, approaching the oracle upper bound. This demonstrates that explicit visual refinement prior to captioning is essential for preserving semantic fidelity in text-mediated retrieval pipelines.

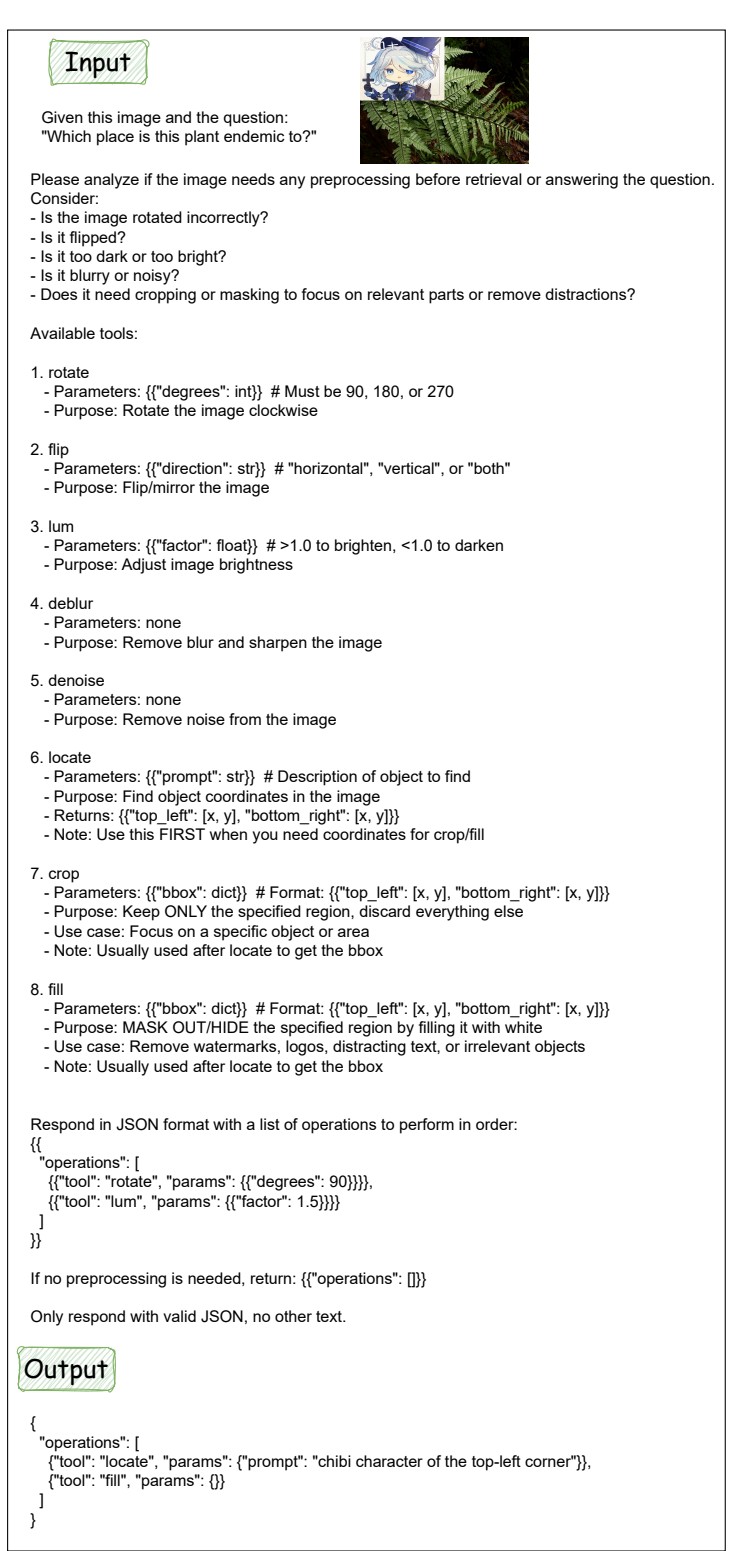

```
Input

Given this image and the question:
"Which place is this plant endemic to?"

Please analyze if the image needs any preprocessing before retrieval or answering the question.
Consider:
- Is the image rotated incorrectly?
- Is it flipped?
- Is it too dark or too bright?
- Is it blurry or noisy?
- Does it need cropping or masking to focus on relevant parts or remove distractions?

Available tools:

1. rotate
   - Parameters: {{"degrees": int}}  # Must be 90, 180, or 270
   - Purpose: Rotate the image clockwise

2. flip
   - Parameters: {{"direction": str}}  # "horizontal", "vertical", or "both"
   - Purpose: Flip/mirror the image

3. lum
   - Parameters: {{"factor": float}}  # >1.0 to brighten, <1.0 to darken
   - Purpose: Adjust image brightness

4. deblur
   - Parameters: none
   - Purpose: Remove blur and sharpen the image

5. denoise
   - Parameters: none
   - Purpose: Remove noise from the image

6. locate
   - Parameters: {{"prompt": str}}  # Description of object to find
   - Purpose: Find object coordinates in the image
   - Returns: {{"top_left": [x, y], "bottom_right": [x, y]}}
   - Note: Use this FIRST when you need coordinates for crop/fill

7. crop
   - Parameters: {{"bbox": dict}}  # Format: {{"top_left": [x, y], "bottom_right": [x, y]}}
   - Purpose: Keep ONLY the specified region, discard everything else
   - Use case: Focus on a specific object or area
   - Note: Usually used after locate to get the bbox

8. fill
   - Parameters: {{"bbox": dict}}  # Format: {{"top_left": [x, y], "bottom_right": [x, y]}}
   - Purpose: MASK OUT/HIDE the specified region by filling it with white
   - Use case: Remove watermarks, logos, distracting text, or irrelevant objects
   - Note: Usually used after locate to get the bbox

Respond in JSON format with a list of operations to perform in order:
{{
  "operations": [
    {{"tool": "rotate", "params": {{"degrees": 90}}}},
    {{"tool": "lum", "params": {{"factor": 1.5}}}}
  ]
}}

If no preprocessing is needed, return: {{"operations": []}}

Only respond with valid JSON, no other text.
```

```
Output

{
  "operations": [
    {"tool": "locate", "params": {"prompt": "chibi character of the top-left corner"}},
    {"tool": "fill", "params": {}}
  ]
}
```

*Figure 15.* Training sample structure for V-QPP(SFT). **Input**: the multimodal prompt consisting of the imperfect query image $I_{query}$ and the instruction template from Table 3. **Output**: the target JSON-formatted operation sequence that the model learns to generate; the loss is computed directly on this predicted sequence against the oracle ground truth.

**Image-to-Image Retrieval (I2I).** We evaluate pure visual matching using the ViQuAE dataset (Lerner et al., 2022), randomly sampling 500 queries from the full dataset. During indexing, only the visual components (Wikipedia images) are embedded using CLIP-ViT-B/32 (Radford et al., 2021); textual descriptions are excluded to enforce vision-only matching. Queries are similarly encoded as visual embeddings for nearest-neighbor search. As shown in the second row of Figure 17, this paradigm exhibits relatively low overall recall, primarily due to the inherent challenges of pure visual matching: without textual semantic guidance, aligning fine-grained semantic concepts based solely on visual similarity becomes difficult, and visual ambiguity is further exacerbated when scaling to large knowledge bases. Nevertheless, clear degradation patterns persist: imperfect queries consistently cause significant recall drops. While zero-shot V-QPP agents yield only marginal improvements, SFT-trained agents achieve substantial gains.

**Composed Retrieval (I+T → I+T).** This paradigm supports compositional queries by jointly encoding image-text pairs. We sample the same 500 examples as Image-to-Image Retrieval paradigm, but now index both the image and its associated Wikipedia article title as a multimodal unit using GME (Zhang et al., 2024). Queries comprising the image and original question are similarly encoded for joint retrieval. As shown in the third row of Figure 17, this paradigm achieves the highest overall recall among all evaluated settings, benefiting from the complementary signals of visual appearance and textual context during matching. The results align with our core findings: despite architectural differences, the *relative degradation patterns* remain consistent—imperfect queries cause significant drops, oracle operations recover most performance, and SFT-trained agents substantially outperform zero-shot baselines—validating V-QPP as a paradigm-agnostic necessity for robust MRAG systems.

**Visual Search via API.** We evaluate MRAG performance under a realistic web-scale retrieval scenario using the Google Lens API, which performs semantic parsing of input images and returns webpages most relevant to the visual content. For each query, we extract titles from the top-5 retrieved webpages as context passages and measure end-to-end question-answering accuracy. Due to the inherent ambiguity in mapping web search results to ground-truth entities, we report only final answer accuracy rather than intermediate retrieval metrics. On a random subset of 100 samples from InfoSeek, results in Table 6 reveal three key patterns: (1) Visual imperfections consistently degrade performance relative to original queries ($0.59 \rightarrow 0.10$–$0.58$), confirming the fragility of API-based retrieval; Figure 16 illustrates a concrete failure case where a simple $180°$ rotation causes Google Lens to completely miss the target entity and return inaccurate results; (2) Oracle pre-processing substantially recovers accuracy—particularly for semantically disruptive perturbations like *Expand* ($0.28 \rightarrow 0.70$), *Watermark* ($0.14 \rightarrow 0.62$), and *RealWorld* ($0.10 \rightarrow 0.40$)—validating V-QPP as a critical restoration mechanism; (3) Our SFT model consistently outperforms zero-shot agents across all perturbation types, demonstrating that learned pre-processing policies can generalize beyond hand-crafted correction rules in open-world search scenarios.

### D.2. Ablation Studies

To verify that our core findings are robust to the choice of embedding model—and not artifacts of a specific retriever architecture—we conduct ablation experiments using three distinct multimodal embedding models:

- **Nomic** (`nomic-embed-text-v1.5`) (Nussbaum et al., 2024): A unified text-image embedding model projecting both modalities into a shared 768-dim space (used as the primary retriever in Section 5.1).
- **GME** (`Alibaba-NLP/gme-Qwen2-VL-2B-Instruct`) (Zhang et al., 2024): A generative multimodal embedding framework optimized for fine-grained visual-text alignment, producing embeddings of dimension **1536**.
- **CLIP** (`openai/clip-vit-base-patch32`) (Radford et al., 2021): The foundational contrastive vision-language pretraining model with a 512-dim embedding space.

For each embedding model, we evaluate five query conditions across Recall@1, Recall@3, and Recall@5: (1) *Original* (canonical $I_{gt}$), (2) *Imperfect* (raw $\tilde{I}$), (3) *Oracle* (perfect tool execution), (4) *V-QPP (Zero-shot)* (off-the-shelf MLLM agent), and (5) *V-QPP (SFT)* (Qwen3-VL-4B-Instruct fine-tuned on our benchmark). Results are shown in Figure 18. Three consistent observations emerge across all embedding models despite variations in absolute recall values: (1) Imperfect queries usually degrade retrieval performance compared to original images, confirming that visual imperfections fundamentally disrupt embedding distributions regardless of the underlying retriever architecture; (2) Oracle tool execution effectively restores performance for most imperfection types—particularly geometric distortions and semantic ambiguities—though limited recovery is observed for irreversible degradations (e.g., Blur, Noise, Crop); (3) Zero-shot V-QPP agents (Non-FT) provide marginal improvements and often fail to generalize across defect types, whereas the SFT-trained agent (V-QPP(SFT)) demonstrates robust cross-retriever generalization, achieving higher recall than zero-shot baselines. This confirms that

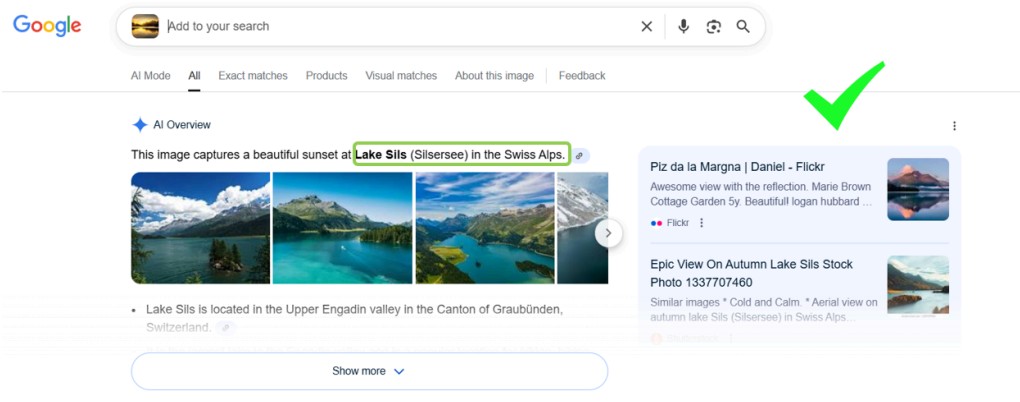

Input the original Image, Google Lens API can give a right answer.

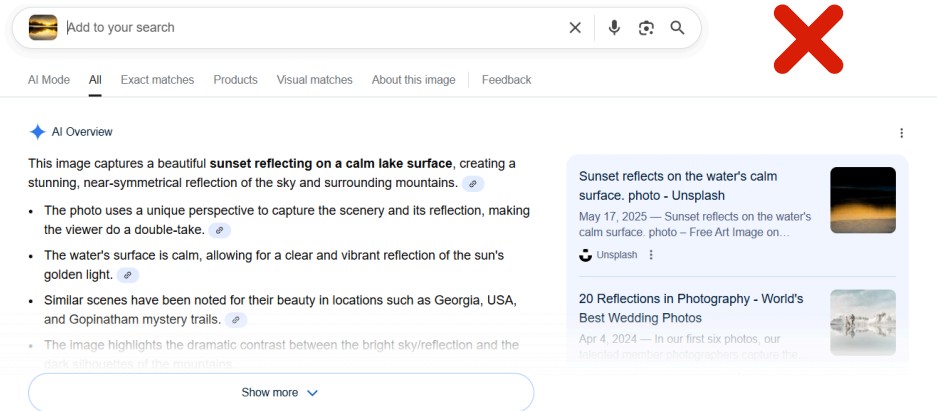

Only Rotate the Image, Google Lens API can NOT give a right answer.

*Figure 16.* Failure case of Google Lens under geometric distortion. A simple $180°$ rotation causes Google Lens to completely miss the target entity.

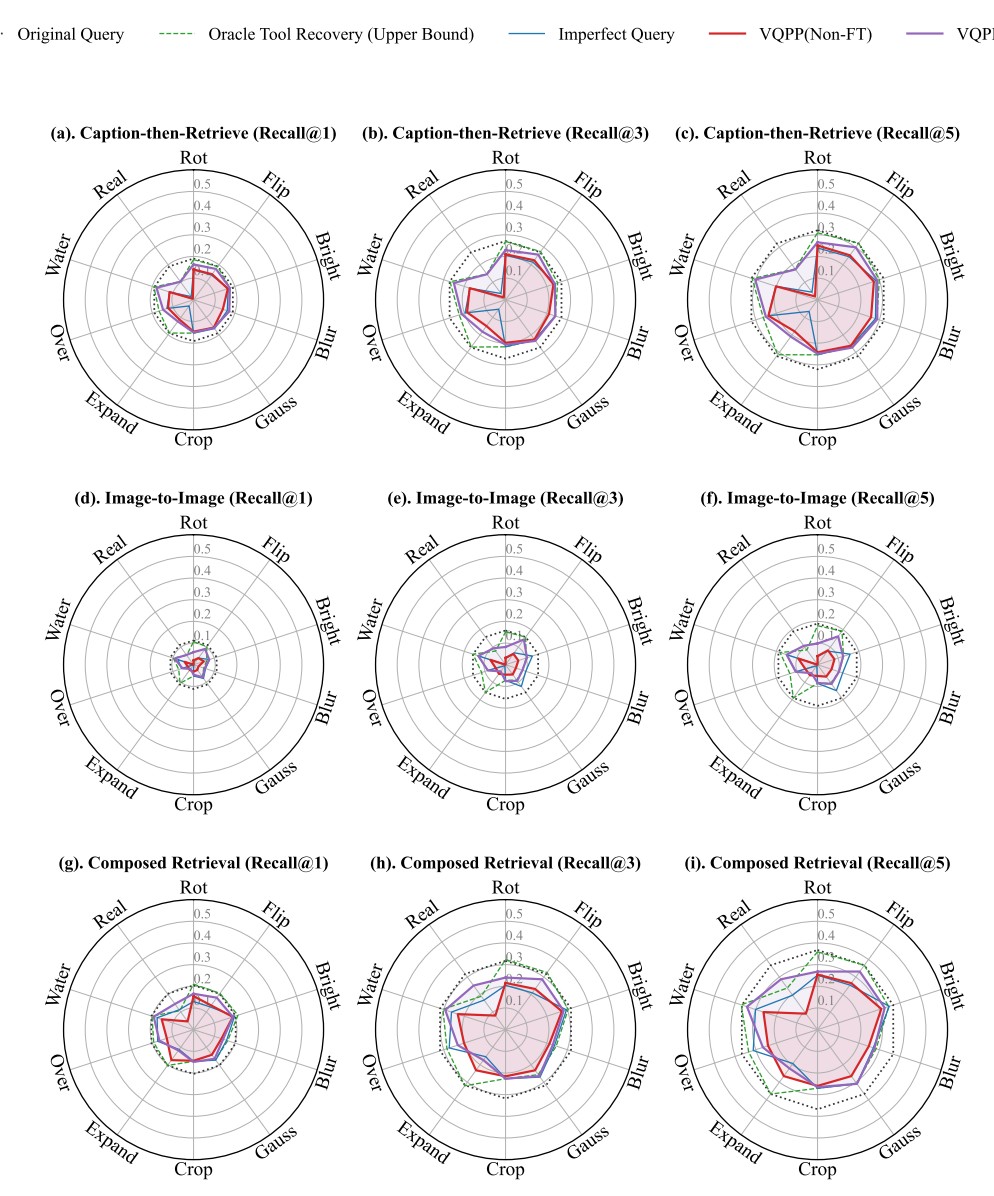

*Figure 17.* Across the Caption-then-Retrieve, Image-to-Image, and Composed Retrieval paradigms, retrieval recall exhibits consistent degradation patterns: despite variations in absolute recall rates due to architectural differences, the *relative degradation patterns* remain highly consistent—imperfect queries consistently cause significant performance drops, oracle operations recover most of the lost performance, and SFT-trained agents substantially outperform zero-shot baselines across paradigms. This cross-paradigm consistency validates visual query pre-processing as a fundamental necessity for MRAG systems, rather than an architecture-specific optimization.

tool-use capability learned on our benchmark transfers effectively to unseen retrieval backends, underscoring the fundamental nature of visual query pre-processing as a retriever-agnostic necessity in MRAG systems.

These results also clarify the relationship between V-QPP and robust retriever design. V-QPP operates on the raw visual query before retrieval, performing pixel-level query refinement analogous to word-level query rewriting in text-based RAG. In contrast, retriever optimization improves the indexing or matching function itself. The consistent improvements from oracle pre-processing and V-QPP(SFT) across Nomic, GME, and CLIP indicate that visual query pre-processing is not tied to a particular retriever architecture. Instead, it is complementary to retriever-side robustness: a stronger retriever may reduce sensitivity to certain imperfections, while V-QPP can further reduce the distribution shift before the query reaches the retriever.

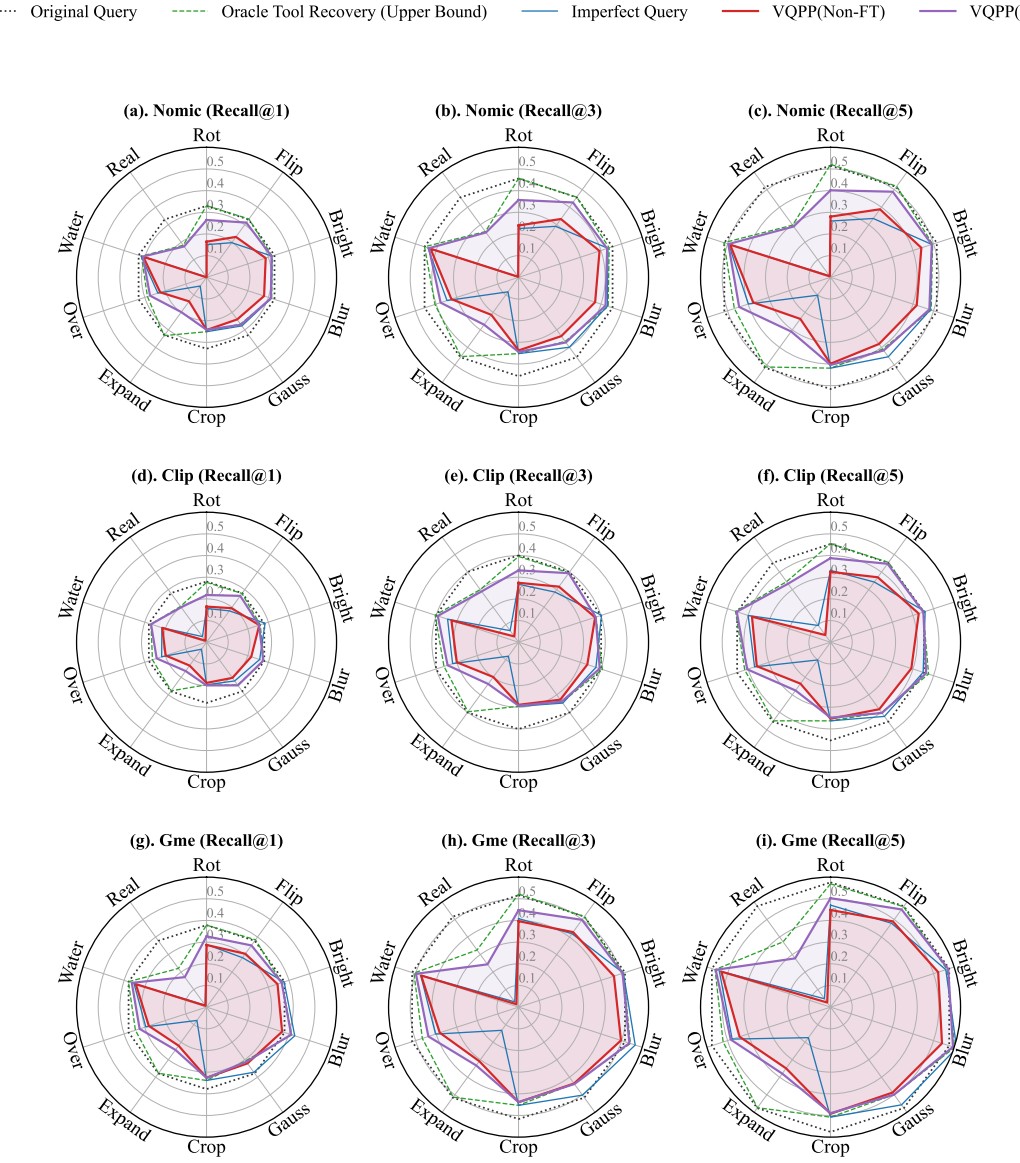

*Figure 18.* Retrieval recall (%) across embedding models and query conditions. Despite variations in absolute recall due to model capacity differences, the *relative degradation patterns* remain consistent: imperfect queries suffer significant performance drops, oracle operations recover most of the lost performance, and SFT-trained agents substantially outperform zero-shot baselines.

## D.3. Generalization to Multi-step Corruptions

Real-world visual queries may contain multiple imperfections simultaneously, such as blur, rotation, and background clutter. To evaluate whether V-QPP agents can generalize beyond isolated single-step corruptions, we construct an additional multi-step corruption split using 1,028 images. For each image, we randomly sample and sequentially apply $n$ corruption operations from the V-QPP taxonomy, where $n \in \{1, 2, 3, 4, 5\}$. We evaluate four settings: directly using the imperfect query, applying oracle preprocessing, using the off-the-shelf V-QPP agent without SFT, and using V-QPP(SFT).

For multi-step corruptions, we do not use TSA as the primary evaluation metric. If TSA were extended to this setting, it could evaluate whether the correct set of tool types is selected independent of execution order. However, such a diagnostic would not fully capture the functional effect of a predicted tool sequence, because different execution orders and parameter choices may lead to different refined images even when the selected tool types are the same. Fully modeling these dependencies is non-trivial and remains an open problem for evaluating agentic visual pre-processing. Therefore, for multi-step evaluation, we primarily rely on retrieval Recall@3 or end-to-end MRAG accuracy, which directly measure whether the generated preprocessing sequence improves downstream retrieval or answer generation.

As shown in Table 7, we report average Recall@3 as a retrieval metric. Retrieval performance decreases monotonically as the number of corruption steps increases, confirming that compositional imperfections introduce substantially greater challenges. The oracle results remain consistently higher than the imperfect-query baseline, indicating that a considerable portion of the retrieval loss is still recoverable through appropriate preprocessing. However, the off-the-shelf V-QPP agent provides only marginal improvements and even slightly underperforms the imperfect baseline when $n = 1$, suggesting that multi-step refinement is difficult without task-specific training. In contrast, V-QPP(SFT) consistently improves over the imperfect baseline across all corruption depths, despite being trained only on single-step corruptions. This demonstrates meaningful generalization from single-step supervision to unseen compositional corruptions, while the remaining gap to the oracle indicates room for future work on multi-step planning and sequential tool-use training.

## D.4. Out-of-distribution Generalization

To further examine whether V-QPP agents generalize beyond the synthetic configurations used in our benchmark, we conduct two out-of-distribution (OOD) evaluations: real-world egocentric images from CRAG-MM and unseen synthetic corruption configurations.

**Real-world egocentric images.** We extend our evaluation with 300 egocentric images from CRAG-MM (Wang et al., 2025b) as imperfect visual queries. For each image, we manually crop the target entity as the oracle query and evaluate retrieval performance using Recall@K. As shown in Table 8, oracle queries substantially improve retrieval over the raw imperfect queries, confirming that real-world egocentric images contain visual noise that can hinder retrieval. The off-the-shelf V-QPP agent underperforms the imperfect baseline, indicating that real-world refinement remains challenging without task-specific training. In contrast, V-QPP(SFT) consistently improves over the imperfect baseline from Recall@1 to Recall@5, suggesting that the tool-use ability learned from V-QPP-Bench transfers positively to real-world egocentric settings.

**Unseen synthetic configurations.** We also evaluate on 1,028 samples with unseen synthetic configurations, including arbitrary rotations from $0°$ to $360°$, varied watermark types and locations, and photo-album-style expand layouts with different numbers of images. As shown in Table 9 and 10, V-QPP(SFT) improves over both the imperfect-query baseline and the off-the-shelf V-QPP agent in most settings. The gains are especially large for unseen expand layouts, where SFT substantially improves both end-to-end accuracy and Recall@3. These results suggest that V-QPP(SFT) does not merely memorize the original synthetic pipeline, but learns transferable visual query preprocessing behaviors. Meanwhile, the remaining gap to oracle performance indicates that OOD visual refinement remains an open challenge.

## D.5. Preliminary Reinforcement Learning Results

Our primary contribution is V-QPP-Bench and a simple SFT baseline for visual query preprocessing. We nevertheless provide a preliminary exploration of reinforcement learning by training with Group Relative Policy Optimization (GRPO) for 0.5 epochs, using retrieval performance as the reward signal. As shown in Table 11, GRPO slightly improves the average Recall@3 over V-QPP(SFT), increasing it from 0.376 to 0.382. The gains are most visible on Flip, Expand, and RealWorld, while improvements are modest or saturated on easier perturbations such as Brightness, Blur, and Watermark. These results

suggest that reinforcement learning may further improve V-QPP agents, but we leave systematic reward design, multi-round rollout, and comparisons among reinforcement learning algorithms to future work.

### D.6. Adaptive Retrieval Evaluation

Our main experiments adopt a forced-retrieval protocol, where the refined visual query is always submitted to the retriever. This design isolates the effect of visual query preprocessing from the additional confounding factor of retrieval-decision making. To evaluate a more deployment-like setting, we further consider an adaptive retrieval protocol, where the agent first decides whether external retrieval is needed before answer generation.

As shown in Table 12, adaptive retrieval yields highly consistent results with forced retrieval. The average absolute difference between the two settings is only about 0.003, and the largest difference is below 0.01. This suggests that most questions in our benchmark require external knowledge and therefore trigger retrieval under the adaptive setting. More importantly, the relative trends remain unchanged: oracle preprocessing provides a strong upper bound, off-the-shelf V-QPP brings limited gains, and V-QPP(SFT) consistently improves over its non-finetuned counterpart across most perturbation types.

### D.7. Latency Analysis

To quantify the runtime cost introduced by the V-QPP agent, we measure latency on 100 randomly selected source images under 11 query conditions, including the original query condition and 10 imperfect query types, resulting in 1,100 samples in total. We decompose the end-to-end runtime according to the three-stage MRAG pipeline: visual query pre-processing, retrieval, and answer generation.

As shown in Table 13, the V-QPP pre-processing stage accounts for 52.2% of the total latency. This stage is dominated by VLM-based operation generation (46.8%), while actual tool invocation only contributes 5.4% of the total latency, with a tool invocation rate of 72.4%. Retrieval accounts for 23.5% of the latency, including VLM-based retrieval decision making (21.7%) and retrieval tool invocation (1.8%), with a retrieval rate of 92.0%. Answer generation accounts for the remaining 24.3%.

Compared with the retrieval-and-generation pipeline without V-QPP pre-processing, the V-QPP stage introduces a relative latency overhead of 109.4%. This overhead mainly comes from the additional VLM call used for operation generation rather than from the image-processing tools themselves. Such latency–quality trade-off is common in agentic retrieval and search frameworks, where additional reasoning and tool-use steps are introduced to improve downstream retrieval reliability (Jin et al., 2025). In our setting, this cost is justified by the substantial retrieval and end-to-end gains brought by V-QPP, especially for severe geometric distortions and semantic ambiguities. Future work may further reduce this overhead through lightweight routing models, cached operation policies, or early-exit mechanisms for visually clean queries.

### D.8. Oracle Decline and Tool-induced Artifacts

Oracle processing provides an upper-bound estimate of V-QPP restoration by applying the correct tools and parameters. However, correct tool invocation does not guarantee monotonic improvement for every individual sample. To quantify this effect, we measure the proportion of samples whose retrieval performance decreases after oracle processing compared with their imperfect-query counterparts. Specifically, we count a sample as declined if the ground-truth evidence is retrieved by the imperfect query under Recall@3 but is no longer retrieved after oracle processing.

As shown in Table 14, only a small fraction of samples exhibit such retrieval decline. The effect is most pronounced for Gaussian noise removal, watermark removal, and deblurring. This is likely because restoration tools can introduce artifacts or alter visual details that affect embedding alignment. For example, denoising may remove fine-grained cues, deblurring may introduce ringing artifacts, and watermark removal or filling may change local texture patterns. These results suggest that traditional CV tools are not always retrieval-neutral, even when invoked correctly. Therefore, oracle results should be interpreted as a practical restoration upper bound under the available tool library, rather than a guarantee of per-sample improvement.

*Table 3.* V-QPP prompt template for visual query pre-processing. The prompt instructs MLLMs to diagnose visual imperfections in $I_{query}$ and output a JSON-formatted operation sequence for tool execution.

---

**Prompt**

Given this image and the question: <image> "<question>"

Please analyze if the image needs any preprocessing before retrieval or answering the question. Consider:
- Is the image rotated incorrectly?
- Is it flipped?
- Is it too dark or too bright?
- Is it blurry or noisy?
- Does it need cropping or masking to focus on relevant parts or remove distractions?

Available tools:

1. rotate
- Parameters: {"degrees": int} # Must be 90, 180, or 270
- Purpose: Rotate the image clockwise

2. flip
- Parameters: {"direction": str} # "horizontal", "vertical", or "both"
- Purpose: Flip/mirror the image

3. lum
- Parameters: {"factor": float} # >1.0 to brighten, <1.0 to darken
- Purpose: Adjust image brightness

4. deblur
- Parameters: none
- Purpose: Remove blur and sharpen the image

5. denoise
- Parameters: none
- Purpose: Remove noise from the image
6. locate - Parameters: {"prompt": str} # Description of object to find
- Purpose: Find object coordinates in the image
- Returns: {"top_left": [x, y], "bottom_right": [x, y]}
- Note: Use this FIRST when you need coordinates for crop/fill

7. crop
- Parameters: {"bbox": dict} # Format: {"top_left": [x, y], "bottom_right": [x, y]}
- Purpose: Keep ONLY the specified region, discard everything else
- Use case: Focus on a specific object or area
- Note: Usually used after locate to get the bbox

8. fill
- Parameters: {"bbox": dict} # Format: {"top_left": [x, y], "bottom_right": [x, y]}
- Purpose: MASK OUT/HIDE the specified region by filling it with white
- Use case: Remove watermarks, logos, distracting text, or irrelevant objects
- Note: Usually used after locate to get the bbox

Respond in JSON format with a list of operations to perform in order:
```
{
  "operations": [
    {"tool": "rotate", "params": {"degrees": 90}},
    {"tool": "lum", "params": {"factor": 1.5}}
  ]
}
```
If no preprocessing is needed, return: {"operations": []}

Only respond with valid JSON, no other text.

---

*Table 4.* Retrieval-decision prompt template. Given the refined image $I'$ and question $Q$, the prompt elicits a binary decision on whether external knowledge retrieval is required.

---

**Prompt**

Given this image and the question: "{question}"

Can you answer this question directly from the image, or do you need additional external knowledge/context?

Respond in JSON format:
```
{
  "need_retrieval": true/false,
  "reason": "brief explanation"
}
```

Only respond with valid JSON, no other text.

---

*Table 5.* Prompt templates for answer generation in the MRAG pipeline.

| **Without Retrieved Context** |
|---|

Question: {question} <image>

Please answer based on the image. Please directly answer the question.

| **With Retrieved Context** |
|---|

Reference Context:
Context 1:
title: {title_1}
text: {text_1}

Context 2:
title: {title_2}
text: {text_2}
. . .

Context k:
title: {title_k}
text: {text_k}

Question: {question} <image>

Please answer the question based on the image and the provided context. Please directly answer the question.

*Table 6.* End-to-end QA accuracy (%) using Google Lens API retrieval on 100 InfoSeek samples. Columns show performance under imperfect queries (no pre-processing), oracle pre-processing (ideal correction), zero-shot MLLM agent (no training), and our SFT agent. Baseline accuracy on original canonical queries: 64%.

| **Perturbation** | **Imperfect** | **Oracle** | **Zero-shot** | **SFT (Ours)** |
|---|---|---|---|---|
| Rotation | 42 | 60 | 40 | 54 |
| Flip | 44 | 62 | 46 | 60 |
| Brightness | 58 | 56 | 44 | 56 |
| Blur | 56 | 52 | 42 | 54 |
| Gaussian Noise | 52 | 54 | 46 | 62 |
| Crop | 50 | 50 | 46 | 50 |
| Expand | 28 | 70 | 34 | 42 |
| Overlay | 54 | 56 | 44 | 48 |
| Watermark | 14 | 62 | 08 | **66** |
| RealWorld | 10 | 40 | 00 | 40 |
| **Average** | 40.8 | 56.2 | 35.0 | 53.0 |

*Table 7.* Recall@3 under multi-step corruptions.

| Step ($n$) | Imperfect | Oracle | V-QPP w/o SFT | V-QPP(SFT) |
|---|---|---|---|---|
| 1 | 0.323 | 0.384 | 0.310 | 0.368 |
| 2 | 0.227 | 0.328 | 0.234 | 0.284 |
| 3 | 0.150 | 0.284 | 0.152 | 0.200 |
| 4 | 0.091 | 0.226 | 0.098 | 0.142 |
| 5 | 0.056 | 0.180 | 0.062 | 0.097 |

*Table 8.* Recall@K on real-world egocentric images from CRAG-MM.

| Recall@K | Imperfect | V-QPP w/o SFT | V-QPP(SFT) | Oracle |
|---|---|---|---|---|
| R@1 | 0.142 | 0.116 | 0.165 | 0.221 |
| R@2 | 0.195 | 0.150 | 0.221 | 0.292 |
| R@3 | 0.206 | 0.157 | 0.236 | 0.326 |
| R@4 | 0.243 | 0.180 | 0.258 | 0.360 |
| R@5 | 0.270 | 0.199 | 0.277 | 0.386 |

*Table 9.* End-to-end accuracy on unseen synthetic corruption configurations.

| OOD Type | Imperfect | V-QPP w/o SFT | V-QPP(SFT) | Oracle |
|---|---|---|---|---|
| Rotation (0°–360°) | 0.304 | 0.285 | 0.320 | 0.415 |
| Watermark | 0.178 | 0.179 | 0.254 | 0.371 |
| Expand | 0.234 | 0.231 | 0.437 | 0.460 |

*Table 10.* Recall@3 on unseen synthetic corruption configurations.

| OOD Type | Imperfect | V-QPP w/o SFT | V-QPP(SFT) | Oracle |
|---|---|---|---|---|
| Rotation (0°–360°) | 0.209 | 0.202 | 0.231 | 0.379 |
| Watermark | 0.397 | 0.382 | 0.410 | 0.411 |
| Expand | 0.047 | 0.050 | 0.208 | 0.310 |

*Table 11.* Preliminary GRPO results measured by average Recall@3.

| Perturbation | Imperfect | Oracle | w/o SFT | SFT | GRPO |
|---|---|---|---|---|---|
| Original | 0.456 | 0.456 | 0.419 | 0.441 | 0.456 |
| Rotation | 0.225 | 0.456 | 0.239 | 0.357 | 0.351 |
| Flip | 0.291 | 0.457 | 0.333 | 0.427 | 0.438 |
| Brightness | 0.434 | 0.440 | 0.393 | 0.430 | 0.434 |
| Blur | 0.431 | 0.435 | 0.372 | 0.422 | 0.430 |
| Noise | 0.399 | 0.366 | 0.336 | 0.372 | 0.373 |
| Crop | 0.352 | 0.352 | 0.338 | 0.344 | 0.354 |
| Expand | 0.083 | 0.452 | 0.214 | 0.270 | 0.290 |
| Overlay | 0.347 | 0.406 | 0.326 | 0.378 | 0.378 |
| Watermark | 0.440 | 0.456 | 0.427 | 0.435 | 0.439 |
| RealWorld | 0.003 | 0.261 | 0.003 | 0.256 | 0.262 |
| Average | 0.315 | 0.412 | 0.309 | 0.376 | 0.382 |

*Table 12.* End-to-end accuracy under forced and adaptive retrieval settings. Each cell reports **Forced / Adaptive** retrieval accuracy.

| Perturbation | Imperfect | Oracle | V-QPP w/o SFT | V-QPP(SFT) |
|---|---|---|---|---|
| Rotation | 0.311 / 0.308 | 0.489 / 0.485 | 0.332 / 0.328 | 0.400 / 0.397 |
| Flip | 0.362 / 0.361 | 0.489 / 0.482 | 0.395 / 0.393 | 0.464 / 0.458 |
| Brightness | 0.462 / 0.458 | 0.464 / 0.461 | 0.423 / 0.420 | 0.457 / 0.452 |
| Blur | 0.433 / 0.431 | 0.434 / 0.431 | 0.393 / 0.390 | 0.436 / 0.430 |
| Gaussian | 0.412 / 0.411 | 0.409 / 0.408 | 0.384 / 0.381 | 0.404 / 0.401 |
| Crop | 0.405 / 0.402 | 0.405 / 0.402 | 0.379 / 0.375 | 0.391 / 0.386 |
| Expand | 0.241 / 0.239 | 0.485 / 0.482 | 0.263 / 0.259 | 0.329 / 0.326 |
| Overlay | 0.393 / 0.391 | 0.464 / 0.461 | 0.369 / 0.366 | 0.440 / 0.437 |
| Watermark | 0.115 / 0.111 | 0.477 / 0.471 | 0.110 / 0.105 | 0.436 / 0.431 |
| RealWorld | 0.160 / 0.160 | 0.319 / 0.317 | 0.047 / 0.048 | 0.287 / 0.285 |

*Note:* Values are reported as **forced retrieval / adaptive retrieval**.

*Table 13.* Latency breakdown of the V-QPP-based MRAG pipeline. Percentages are averaged over 1,100 samples.

| Stage | Latency Share | Rate |
|---|---|---|
| V-QPP pre-processing | 52.2% | 72.4% |
| Retrieval | 23.5% | 92.0% |
| Answer generation | 24.3% | – |

*Table 14.* Proportion of samples with retrieval decline after correct tool invocation. A decline means that Recall@3 succeeds for the imperfect query but fails after oracle processing.

| Operation | Declined |
|---|---|
| Gaussian noise | 6.1% |
| Watermark | 4.1% |
| Blur | 3.7% |
| Brightness | 3.2% |
| Overlay | 2.4% |
| Rotation | 1.7% |
| Flip | 1.7% |
| Expand | 0.6% |
| RealWorld | 0.3% |

