# OpenReview forum: "Fix Before Search: Benchmarking Agentic Visual Query Pre-processing in Multimodal Retrieval-augmented Generation"
_ICML.cc/2026/Conference — ICML 2026 regular_

### Official Review · Reviewer_N48d · 2026-03-11

**Soundness:** 3
**Presentation:** 3
**Significance:** 3
**Originality:** 2
**Overall Recommendation:** 4
**Confidence:** 4

**Summary:**

This paper introduces V-QPP-Bench, a benchmark for evaluating visual query pre-processing in Multimodal RAG—a stage analogous to query rewriting in text RAG but largely neglected in the visual domain. The benchmark contains 46,700 corrupted queries across 10 imperfection types, framing the correction task as agentic tool-use. Furthermore, the authors show that supervised fine-tuning on just 1,000 examples enables Qwen3-VL-4B to rival or surpass larger proprietary models, suggesting that V-QPP is a learnable capability and that the benchmark can serve as an effective training resource for building more robust MRAG systems.

**Compliance With Llm Reviewing Policy:**

Affirmed.

**Final Justification:**

The supplementary experiments and clarifications have sufficiently addressed my key concerns.

**Key Questions For Authors:**

1. Imperfect images may retain robust visual features useful for retrieval. Among oracle-processed samples, do any show worse performance than their imperfect counterparts? If so, what proportion, and are they concentrated in specific imperfection types or image categories?

2. What is the average latency overhead introduced by the V-QPP agent per query, including MLLM diagnosis and tool execution? How does this compare to the retrieval and generation stages?

3. Has the SFT-trained model been evaluated on genuinely imperfect real-world images rather than synthetic corruptions? Even a small-scale evaluation would clarify whether the model learns general visual diagnostic capabilities or merely memorizes synthetic corruption patterns.

4. Can you provide results on queries with multiple concurrent imperfections? How does the SFT model handle compositions not seen during training? This would validate whether the model has genuine multi-step reasoning ability.

**Limitations:**

Yes

**Strengths And Weaknesses:**

Strengths:
1. The paper identifies a genuine and underexplored gap, namely, visual query preprocessing in MRAG, and frames it as an agentic tool-use task. This formulation is well-motivated and connects naturally to the broader trend of tool-augmented LLMs.

2. The "reverse-engineering" data construction paradigm is principled, enabling multi-level evaluation across tool selection, parameter prediction, retrieval recall, and end-to-end accuracy. The experimental coverage is extensive.

3. The paper is clearly written with effective visualizations and illustrative case studies in the appendix that build good intuition for the problem.

Weaknesses:
1. Imperfect images may still contain robust visual features that retrievers can leverage for matching. Preprocessing operations like deblurring or denoising, even when correctly applied, may destroy such features. Section 5.3 acknowledges limited oracle improvement for quality degradations but does not examine whether oracle-processed samples can actually perform worse than their imperfect counterparts.

2. Section 3.2 models imperfections as compositions of multiple degradation functions, yet the benchmark evaluates only single-corruption scenarios. This gap between formulation and evaluation weakens the claims about agentic decision-making, as compositional corruptions require multi-step reasoning that single corruptions do not.

---

> ### Author Rebuttal · Authors · 2026-03-30
>
> Dear Reviewer N48d,
>
> We thank you for your constructive feedback. We hope our responses address your concerns and merit a higher score.
>
> > Q1: Do any oracle-processed samples perform worse than their imperfect counterparts?
>
> **A1:** Thanks for your insightful question. As shown in Table 1, even with correct tool invocation, a subset of samples still sees retrieval decline — most pronounced for Gaussian noise removal, watermark removal, and deblurring (6.1%, 4.1%, and 3.7% respectively). This likely stems from partial irreversibility: restoration tools can introduce their own artifacts (e.g., ringing effects from deblurring, color distortion from brightness correction) that disrupt embedding alignment more severely than the original degradation. We will include a discussion on the potential reasons behind this finding in our revision.
>
> **Table 1: Proportion of Samples with Retrieval Decline after Correct Tool Invocation (Recall@3)**
> | | Gaussian | Watermark | Blur | Brightness | Overlay | Rotation | Flip | Expand | RealWorld |
> |-|-|-|-|-|-|-|-|-|-|
> | Declined (↓) | 6.1% | 4.1% | 3.7% | 3.2% | 2.4% | 1.7% | 1.7% | 0.6% | 0.3% |
>
> ---
>
> > Q2: Does V-QPP support multi-step corruption evaluation? How does the SFT model handle compositions not seen during training?
>
> **A2:**
>
> **Multi-Step Corruption Results:** We evaluate on 1,028 images with $n$ corruption steps (1–5, random types) to better simulate real-world complexity, reporting average Recall@3 in Table 2.
>
> **Table 2: Recall@3 under Multi-Step ($n$) Corruption**
> |Step($n$)| Imperfect |Oracle| V-QPP(W/O SFT) |V-QPP(SFT)|
> |-|-|-|-|-|
> |1| 0.323 |0.384| 0.310 |0.368|
> |3| 0.150 |0.284| 0.152 |0.200|
> |5| 0.056 |0.180| 0.062 |0.097|
>
> ($n$=2,4 will be included in revision.) Performance degrades as $n$ increases, confirming greater challenges under multi-step corruption. Notably, our SFT model — trained only on single-step corruptions — still meaningfully improves retrieval, demonstrating strong generalization to unseen compositional corruptions. Manual inspection further reveals that the model learns to prioritize the most severe corruption for repair. These findings and construction guidelines for multi-step corruption data will be incorporated in our revision.
>
> **Advanced Multi-Step Directions:** We clarify that our contribution lies in establishing the benchmark and a simple yet effective SFT baseline; developing advanced multi-step training methods is beyond our current scope. Nevertheless, our benchmark and synthetic pipeline serve as an ideal testbed for such exploration. Promising directions include: (1) leveraging our pipeline to automatically construct multi-step trajectories for SFT, and (2) multi-round RL rollout with retrieval performance as reward — both readily built upon our existing framework.
>
> ---
>
> > Q3: Has the SFT-trained model been evaluated on genuinely imperfect real-world images?
>
> **A3:** Following your suggestion, we extend V-QPP-Bench with 300 egocentric images from CRAG-MM [1] as imperfect queries, with manually cropped ground-truth entities as oracle queries, reporting Recall@K in Table 3.
>
> **Table 3: Recall@K on Real-World Egocentric Images (CRAG-MM)**
> | Recall@K | Imperfect | V-QPP(W/O SFT) | V-QPP(SFT) | Oracle |
> |-|-|-|-|-|
> | 1 | 0.142 | 0.116 | 0.165 | 0.221 |
> | 2 | 0.195 | 0.150 | 0.221 | 0.292 |
> | 3 | 0.206 | 0.157 | 0.236 | 0.326 |
> | 4 | 0.243 | 0.180 | 0.258 | 0.360 |
> | 5 | 0.270 | 0.199 | 0.277 | 0.386 |
>
> Results confirm: (a) imperfect images significantly degrade retrieval performance; (b) oracle processing yields substantial gains, yet current MLLMs remain limited; (c) the same SFT model trained on our V-QPP data transfers effectively to real-world settings, validating the generalizability of our synthetic training pipeline. These results will be incorporated in our revision.
>
> ---
>
> > Q4: What is the average latency overhead introduced by the V-QPP agent per query?
>
> **A4:** We evaluate latency on 100 randomly selected images across 11 imperfect operation types (1,100 samples total), broken down as follows:
>
> - **Image preprocessing (V-QPP):** 52.2% — VLM-based operation generation (46.8%) + tool invocation (5.4%), with a tool invocation rate of 72.4%
> - **Retrieval:** 23.5% — VLM-based retrieval decision (21.7%) + retrieval tool invocation (1.8%), with a retrieval rate of 92%
> - **Answer generation:** 24.3%
>
> The V-QPP preprocessing stage introduces a latency overhead of 109.4% relative to the retrieval-and-generation pipeline alone. We note that such overhead is inherent to agentic search frameworks in general — e.g., Search-R1 [2] — where iterative reasoning and tool invocation necessarily trade latency for improved performance, and is consistent with the cost observed in typical agentic search systems.
>
> ---
>
> [1] CRAG-MM: Multi-modal Multi-turn Comprehensive RAG Benchmark.
>
> [2] Search-R1: Training LLMs to Reason and Leverage Search Engines with Reinforcement Learning.

---

> > ### Author Rebuttal · Reviewer_N48d · 2026-04-02
> >
> > The supplementary experiments and clarifications have sufficiently addressed my key concerns. I am satisfied with their responses and expect the promised revisions to be included in the final paper.

---

> > > ### Author Response · Authors · 2026-04-02
> > >
> > > Dear Reviewer N48d,
> > >
> > > Thank you for your acknowledgement. We are glad to hear that your concerns are fully resolved.
> > >
> > > As we have addressed all the key concerns, **we would greatly appreciate it if you could consider further improving your score to reflect this positive assessment.** Please let us know if you have any further questions before the discussion ends, and we would be happy to provide more information.
> > >
> > > We will ensure all promised revisions are included in the final paper.
> > >
> > > Best regards,
> > >
> > > Authors

---

### Official Review · Reviewer_EFeg · 2026-03-13

**Soundness:** 3
**Presentation:** 2
**Significance:** 3
**Originality:** 3
**Overall Recommendation:** 4
**Confidence:** 3

**Summary:**

This paper introduces V-QPP-Bench, a benchmark designed to evaluate how visual query imperfections degrade Multimodal Retrieval-Augmented Generation (MRAG) systems and to test the ability of Multimodal Large Language Models (MLLMs) to actively pre-process and restore these queries. The authors systematically categorize 10 types of visual corruptions across geometric, quality, and semantic dimensions, demonstrating that these flaws cause catastrophic retrieval failures. By framing query refinement as an agentic tool-use task, the study reveals that off-the-shelf MLLMs struggle significantly with tool selection and parameter prediction. However, the paper shows that supervised fine-tuning enables smaller models, like a 4B parameter MLLM, to match or exceed proprietary models in recovering retrieval performance. Overall, the work highlights a critical vulnerability in MRAG pipelines and proposes a novel, active-agent framework for visual query refinement.

**Compliance With Llm Reviewing Policy:**

Affirmed.

**Final Justification:**

Rebuttal addressed my main concerns

**Key Questions For Authors:**

1. How does the Tool Selection Accuracy (TSA) metric account for functionally equivalent but differently ordered sequences of tool operations?
2. Could you justify the specific choice of using an exponential similarity metric to evaluate the tool parameter prediction, rather than a simpler continuous error metric?
3. What were the specific inference generation parameters used for the MLLMs across the various evaluation stages?

**Limitations:**

Yes

**Strengths And Weaknesses:**

Strengths:
1. The paper addresses a highly important and practical vulnerability in MRAG systems, successfully demonstrating that assuming canonical, perfect visual inputs is a flawed premise in real-world deployments.
2. Formulating visual query pre-processing as an active, agentic decision-making task where models must diagnose and deploy tools is a strong conceptual contribution.

Weaknesses:
1. Relying entirely on artificially injected synthetic transformations (e.g., exact 90-degree rotations or fixed grid expansions) is a well-established vulnerability in robustness benchmarks, as it rarely mimics the compounded complexities of true "in-the-wild" captures.
2. Adding visual information is not necessarily a distractor and could actually be relevant to the query. This might lead to ambiguity.
3. The Tool Selection Accuracy (TSA) metric appears overly rigid by demanding exact matching against an oracle sequence, failing to account for cases where different permutations of tool calls could achieve the same refined visual outcome.
4. The authors did not provide crucial inference generation parameters such as temperature and top-k.
5. The core fine-tuning baseline relies on a relatively small 4B parameter model learning to reverse-engineer the exact synthetic pipeline used to create the test set, raising concerns about overfitting and true generalization.

---

> ### Author Rebuttal · Authors · 2026-03-30
>
> Dear Reviewer EFeg,
>
> We thank you for your constructive feedback. We hope our responses address your concerns and merit a higher score.
>
> > Q1: Rely on artificially injected synthetic transformations; it rarely mimics the compounded complexities of true "in-the-wild" captures.
>
> **A1:** Our operations are directly inspired by real-world scenarios (Appendix A.1), and we already include real-world screen-captured images (4 types, 1,028 samples). We further address this with two new evaluations:
>
> **Real-World Egocentric Evaluation:** We extend V-QPP-Bench with 300 egocentric images from CRAG-MM [1] as imperfect queries, with manually cropped entities as oracle queries, reporting Recall@3 in Table 1.
>
> **Table 1: Recall@3 on OOD Real-World Egocentric Images (CRAG-MM)**
> |OOD Type| Imperfect |V-QPP(W/O SFT)| V-QPP(SFT) |Oracle|
> |-|-|-|-|-|
> |Real-world egocentric|0.206|0.157|0.236|0.326|
>
> Results confirm: (a) imperfect images degrade retrieval; (b) oracle processing yields substantial gains; (c) SFT transfers effectively to real-world settings.
>
> **Multi-Step Corruption:** We evaluate on 1,028 images with $n$ corruption steps (1–5, random types), reporting average Recall@3 in Table 2.
>
> **Table 2: Recall@3 under Multi-Step ($n$) Corruption**
> |Step($n$)| Imperfect |Oracle| V-QPP(W/O SFT) |V-QPP(SFT)|
> |-|-|-|-|-|
> |1|0.323|0.384|0.310|0.368|
> |3|0.150|0.284|0.152|0.200|
> |5|0.056|0.180|0.062|0.097|
>
> ($n$=2,4 will be included in revision.) Performance degrades as $n$ increases. Notably, our SFT model — trained only on single-step corruptions — still improves retrieval, demonstrating generalization to unseen compositional corruptions.
>
> ---
>
> > Q2: Whether the model merely overfits to the synthetic pipeline rather than learning generalizable preprocessing.
>
> **A2:** We conduct OOD analysis across two unseen settings:
>
> **Unseen Real-World OOD:** As shown in Table 1, V-QPP(SFT) transfers effectively to real-world egocentric images, recovering substantial performance lost to real-world imperfections.
>
> **Unseen Synthetic Configurations:** Table 3 reports results on unseen configurations: random rotation (0–360°), varied watermarks, and photo-album-style expand layouts (1,028 samples).
>
> **Table 3: Final QA Accuracy on OOD Rotation, Watermark and Expand Images**
> |OOD Type| Imperfect |V-QPP(W/O SFT)|V-QPP(SFT) |Oracle|
> |-|-|-|-|-|
> |Rotation (0–360°)|0.304|0.285|0.320|0.415|
> |Watermark (varied types/locations)|0.178|0.179|0.254|0.371|
> |Expand (varied layouts/image numbers)| 0.234 |0.231|0.437 |0.460|
>
> These results demonstrate that SFT equips the model with generalizable visual query preprocessing ability beyond simply reversing the synthetic pipeline.
>
> ---
>
> > Q3: Adding visual information is not necessarily a distractor.
>
> **A3:** We agree that additional visual information can be beneficial in real-world cases — in which case the correct operation is to retain rather than remove it.  However, in our benchmark, Semantic Ambiguity corruptions are constructed by adding irrelevant content (e.g., watermarks, overlay images, grid distractors) that is strictly unhelpful by design, making removal the correct operation. We will add a clarifying discussion in our revision.
>
> ---
>
> > Q4: How does TSA account for functionally equivalent but differently ordered tool sequences?
>
> **A4:** Our benchmark currently focuses on single-step corruptions, where our orthogonal tool design — each tool addressing a distinct degradation type — leaves little room for equivalent substitutions. The one exception — 180° rotation being equivalent to horizontal and vertical flips — is already handled in our calculation.
>
> For multi-step cases, TSA evaluates *whether the correct tools are selected*, independent of execution order. We acknowledge that different orderings and parameters can still lead to different outcomes — fully modeling such dependencies is an open problem. This is why we recommend complementing TSA with Retrieval Performance and End-to-End MRAG metrics.
>
> ---
>
> > Q5: Justification for exponential similarity in tool parameter evaluation.
>
> **A5:** Exponential similarity is used only for the `lum` (brightness) tool. We chose it because it shares the same properties as our other parameter metrics (Mean Accuracy, IoU): bounded in [0, 1], higher is better, approaching 0 for worst-case predictions — ensuring consistent scaling when computing the product with TSA. A simple L2 metric lacks this boundedness and is more sensitive to outliers.
>
> Following your suggestion, we will also report L2-based results in our revision: V-QPP (W/O SFT) achieves 0.620 / 2.037; V-QPP (SFT) achieves 0.884 / 1.740 (exponential / L2). The two metrics are consistent.
>
> ---
>
> > Q6: Specific inference generation parameters.
>
> **A6:** We used greedy decoding (temperature = 0.0), top-p =1.0, retrieve top-k=5, and max tokens=1024, with all other parameters at SGLang defaults. These details will be added in our revision.
>
> ---
>
> [1] CRAG-MM: Multi-modal Multi-turn Comprehensive RAG Benchmark.

---

> > ### Author Rebuttal · Reviewer_EFeg · 2026-04-02
> >
> > The authors addressed my concerns and I am updating my score accordingly

---

> > > ### Author Response · Authors · 2026-04-03
> > >
> > > Dear Reviewer EFeg,
> > >
> > > Thank you very much for your positive feedback and for increasing the score. We are glad that our responses and the additional experiments successfully addressed your concerns. We will ensure all the discussed clarifications and new results are incorporated into the final version of the paper.
> > >
> > > Best regards,
> > >
> > > Authors

---

### Official Review · Reviewer_PDvj · 2026-03-13

**Soundness:** 3
**Presentation:** 3
**Significance:** 3
**Originality:** 3
**Overall Recommendation:** 4
**Confidence:** 4

**Summary:**

This paper discusses the following issue: while text-based RAG routinely employs query rewriting to bridge semantic gaps, current MRAG pipelines predominantly treat visual inputs as static and immutable, leaving them vulnerable to real-world image degradations. To address this, the authors introduce V-QPP-Bench, a benchmark comprising 46,700 imperfect queries to evaluate Visual Query Pre-Processing (V-QPP). By formulating V-QPP as an agentic decision-making task, the paper evaluates the ability of MLLMs to autonomously diagnose visual imperfections and deploy a library of perceptual tools (e.g., cropping, deblurring) to refine queries before retrieval. The results demonstrate that visual imperfections severely degrade MRAG performance, but oracle tool selection can recover accuracy. Furthermore, while off-the-shelf MLLMs struggle with accurate tool selection, supervised fine-tuning (SFT) allows smaller models (e.g., Qwen3-VL-4B) to surpass proprietary models, validating the learnability of the proposed framework.

**Compliance With Llm Reviewing Policy:**

Affirmed.

**Key Questions For Authors:**

Please refer to the weaknesses

**Limitations:**

No. Please add discussions regarding limitations.

**Strengths And Weaknesses:**

[Strengths]

The paper introduces an interesting perspective by formulating visual query pre-processing as an agentic tool-use task.

The construction of V-QPP-Bench seems sound.

The authors validate their findings across five distinct retrieval paradigms (including Text-Targeted, Vision-Inclusive, and Search API Integration) and conduct ablations across three different embedding models (Nomic, GME, CLIP).

[Weaknesses]

I am a little confused by the SFT task formulation regarding generalization. The SFT objective trains the model to predict the deterministic reverse parameters ($f^{-1}$) of 10 hardcoded synthetic operations. Could the authors address whether this evaluates the model's ability to reverse-engineer the specific data-generation script rather than learning generalized visual intent clarification?

Could you explain the rationale behind defining the "Oracle" action solely as pixel-level restoration ($I_{gt}$)? Dense encoders typically rely on semantic features rather than exact pixel fidelity.

I wonder if the authors could elaborate on whether the chosen traditional CV tools (e.g., Richardson-Lucy deconvolution) might introduce high-frequency artifacts that actively disrupt embedding alignment.

A point of clarification is needed regarding the evaluation pipeline in Algorithm 1. The primary evaluation uses a "forced retrieval" protocol, which bypasses the agent's decision on whether retrieval is actually necessary.

It is not entirely clear how the technical implementation of the tool library handles coordinate system shifts during multi-step reasoning ($O = [o_1, o_2, ..., o_k]$). For example, if an agent applies crop or rotate, the absolute pixel coordinates for a subsequent locate or fill operation would fundamentally change.

---

> ### Author Rebuttal · Authors · 2026-03-30
>
> Dear Reviewer PDvj,
>
> We thank you for your constructive feedback. We hope our responses address your concerns and merit a higher score.
>
> > Q1: Formulation & SFT Generalizability: Does SFT merely learn to reverse-engineer the synthetic pipeline rather than acquiring generalizable visual query preprocessing ability?
>
> **A1:** Our tool library consists of general-purpose operations applicable to any image, not a fixed inverse pipeline. We conduct OOD analysis across two unseen settings:
>
> **Unseen Real-World OOD:** We extend V-QPP-Bench with 300 egocentric images from CRAG-MM [1] as imperfect queries and manually cropped entities as oracle queries. As shown in Table 1, V-QPP(SFT) transfers effectively to real-world settings, validating generalizability beyond the synthetic pipeline.
>
> **Unseen Synthetic Configurations:** Table 2 reports results on unseen configurations: random rotation (0–360°), varied watermarks, and photo-album-style expand layouts.
>
> **Table 1: Recall@3 on OOD Real-World Egocentric Images (CRAG-MM)**
> | OOD Type | Imperfect | V-QPP(W/O SFT) | V-QPP(SFT) | Oracle |
> |-|-|-|-|-|
> | Real-world egocentric | 0.206 | 0.157 | 0.236 | 0.326 |
>
> **Table 2: Final QA Accuracy on OOD Rotation, Watermark and Expand Images**
> |OOD Type| Imperfect |V-QPP(W/O SFT)| V-QPP(SFT) |Oracle|
> |-|-|-|-|-|
> |Rotation (0–360°)| 0.304 |0.285| 0.320 |0.415|
> |Watermark (varied types/locations)| 0.178 |0.179| 0.254 |0.371|
> |Expand (varied layouts/image numbers)| 0.234 |0.231| 0.437 |0.460|
>
> **These results demonstrate that SFT equips the model with generalizable preprocessing ability beyond reversing the synthetic pipeline.**
>
> ---
>
> > Q2: Rationale behind defining the "Oracle" action solely as pixel-level restoration.
>
> **A2:** Query preprocessing operates on the **raw query before it enters the retriever** — in the visual setting, this means **pixel-level manipulation only**, directly analogous to **word-level rewriting in text-based RAG** [2][3]. Feature-level intervention would require modifying the retrieval process itself, which falls outside query preprocessing scope.
>
> We also agree feature-level retrieval signals could serve as rewards to guide V-QPP toward better pixel-level processing — a promising future direction.
>
> ---
>
> > Q3: How does V-QPP perform under adaptive (non-forced) retrieval settings?
>
> **A3:** We adopt forced retrieval as our primary protocol to isolate the effect of visual query preprocessing. Results under adaptive retrieval (Table 3) remain consistent, suggesting most questions require external knowledge and trigger retrieval regardless.
>
> **Table 3: End-to-End Accuracy: Forced / Adaptive Retrieval**
> | Perturbation | Imperfect | Oracle | V-QPP(W/O SFT) | V-QPP(SFT) |
> |-|-|-|-|-|
> |Rotation | 0.311 / 0.308 | 0.489 / 0.485 | 0.332 / 0.328 | 0.400 / 0.397 |
> |Flip | 0.362 / 0.361 | 0.489 / 0.482 | 0.395 / 0.393 | 0.464 / 0.458 |
> |Brightness | 0.462 / 0.458 | 0.464 / 0.461 | 0.423 / 0.420 | 0.457 / 0.452 |
> |Blur | 0.433 / 0.431 | 0.434 / 0.431 | 0.393 / 0.390 | 0.436 / 0.430 |
> |Gaussian | 0.412 / 0.411 | 0.409 / 0.408 | 0.384 / 0.381 | 0.404 / 0.401 |
> |Crop | 0.405 / 0.402 | 0.405 / 0.402 | 0.379 / 0.375 | 0.391 / 0.386 |
> |Expand | 0.241 / 0.239 | 0.485 / 0.482 | 0.263 / 0.259 | 0.329 / 0.326 |
> |Overlay | 0.393 / 0.391 | 0.464 / 0.461 | 0.369 / 0.366 | 0.440 / 0.437 |
> |Watermark | 0.115 / 0.111 | 0.477 / 0.471 | 0.110 / 0.105 | 0.436 / 0.431 |
> |RealWorld | 0.160 / 0.160 | 0.319 / 0.317 | 0.047 / 0.048 | 0.287 / 0.285 |
>
> ---
>
> > Q4: Can traditional CV tools introduce artifacts that hurt retrieval performance?
>
> **A4:** As shown in Table 4, even with correct tool invocation, a subset of samples still sees retrieval decline, likely due to introduced artifacts. This effect is most pronounced for Gaussian noise removal, watermark removal, and deblurring (6.1%, 4.1%, and 3.7% respectively). We will discuss the potential reasons in our revision.
>
> **Table 4: Proportion of Samples with Retrieval Decline after Correct Tool Invocation**
> | | Gaussian | Watermark | Blur | Brightness | Overlay | Rotation | Flip | Expand | RealWorld |
> |-|-|-|-|-|-|-|-|-|-|
> | Declined (↓) | 6.1% | 4.1% | 3.7% | 3.2% | 2.4% | 1.7% | 1.7% | 0.6% | 0.3% |
>
> ---
>
> > Q5: How to handle coordinate shift?
>
> **A5:** Only `crop` and `fill` require coordinates, and our prompt design enforces that `locate` is always called immediately before either. Crucially, `locate` runs on the *current* image state — after any prior transformations have been applied — so the returned coordinates are always aligned with the actual pixel content. As long as the agent provides an accurate text prompt to `locate` (e.g., *"the red handbag held by the person"*), coordinate shift is never an issue regardless of how many steps preceded it.
>
> ---
>
> [1] CRAG-MM: Multi-modal Multi-turn Comprehensive RAG Benchmark.
>
> [2] Deepretrieval: Hacking real search engines and retrievers with LLMs via reinforcement learning.
>
> [3] s3: You don't need that much data to train a search agent via rl.

---

> > ### Author Rebuttal · Reviewer_PDvj · 2026-04-06
> >
> > Thanks for the reply. My concerns have been addressed.

---

> > > ### Author Response · Authors · 2026-04-06
> > >
> > > Dear Reviewer PDvj,
> > >
> > > Thank you for your acknowledgement. We are glad to hear that your concerns are fully resolved. Please let us know if you have any further questions before the discussion ends, and we would be happy to provide more information.
> > >
> > > We will ensure all promised revisions are included in the final paper.
> > >
> > > Best regards,
> > >
> > > Authors

---

### Official Review · Reviewer_BFg7 · 2026-03-22

**Soundness:** 2
**Presentation:** 3
**Significance:** 3
**Originality:** 3
**Overall Recommendation:** 4
**Confidence:** 4

**Summary:**

This paper introduces V-QPP-Bench, a benchmark for evaluating visual query preprocessing in Multimodal RAG systems, arguing that imperfect real-world images (rotated, blurry, cluttered) cause catastrophic retrieval failures that current pipelines ignore. The benchmark comprises 46,700 corrupted queries spanning 10 imperfection types, with an agentic framework where MLLMs must diagnose flaws and apply perceptual tools (rotate, crop, deblur, etc.) to fix images before retrieval. Key findings show that imperfections can degrade retrieval recall by up to 98.6%, that oracle preprocessing recovers near-perfect performance while off-the-shelf MLLMs struggle without training, and that fine-tuning a small 4B model on just 1,000 samples enables it to match or exceed GPT-4o at this task.

**Compliance With Llm Reviewing Policy:**

Affirmed.

**Key Questions For Authors:**

- Table 6 (Google Lens) is based on only 25 samples, which is too small for reliable conclusions.
- What happens when the agent encounters a corruption type not seen during training?
- The SFT dataset only includes 100 unique images and 1k trajectories; could it be too small a training set?
- Have you considered using RL further after SFT?
- How do you decide the type of the image corruption operations? Have you observed the typical corruption type from real-world images and gotten the corruption distribution?

**Limitations:**

No. See weakness

**Strengths And Weaknesses:**

**Strengths**

- **Well-motivated and timely problem**. The paper identifies a genuine blind spot in MRAG research. Text RAG has extensively studied query preprocessing, but the visual side has been largely ignored. The analogy between text query rewriting and visual query refinement is compelling and well-articulated.

- **Comprehensive benchmark design**. The taxonomy of 10 imperfection types across three categories is thorough and well-grounded in real-world scenarios (photographing documents at angles, motion blur, cluttered shelves). The reverse-engineering construction paradigm is elegant — corrupting clean images provides automatic oracle ground truth for both tool selection and parameters.

- **Extensive evaluation**.  Strong empirical findings. The vulnerability analysis is striking (98.6% recall drop for RealWorld), and the gap between oracle and zero-shot agent performance clearly demonstrates both the potential and the current limitation. Also, the strong retrieval performance does not secure decent MRAG end-to-end performance, as watermark drags down the end-to-end performance a lot, but the retrieval performance is only marginally affected. The SFT results with only 1,000 training samples are promising and practically useful.

**Weaknesses**
- **Synthetic-only imperfections limit ecological validity**. While the corruptions are motivated by real-world scenarios, they are all synthetically generated with controlled parameters. Real-world imperfections are messier — perspective distortion isn't exactly 90°/180°/270° rotation; blur isn't purely Gaussian; clutter doesn't come in neat 2×2 grids. The "RealWorld" category uses synthetic templates that embed images into screen scenes, which is still quite artificial. The paper acknowledges focusing on single corruptions for fine-grained analysis, but compound corruptions are the norm in practice. This limits the benchmark's claim of evaluating "real-world" robustness.

- **Lacking OOD analysis for SFT**. The image set in the test query is corrupted using the same 3 types of corruption operations, as in the SFT training data. However, for the real-world multimodal query, it may fall outside the distribution induced by the corruption types.mentioned in this paper. Could it overfit the SFT training data, especially considering the watermark performance is boosted after SFT training? Could it be limited to a certain type of watermark in the training data?

- **Missing baselines**. The paper doesn't compare against retrieval-side robustness approaches (e.g., data augmentation during encoder training, robust embedding methods). It would strengthen the paper to show that V-QPP is complementary to or better than making the retriever itself more robust.

---

> ### Author Rebuttal · Authors · 2026-03-30
>
> Dear Reviewer  BFg7,
>
> We thank you for your constructive feedback. We hope our responses address your concerns and merit a higher score.
>
>
> > Q1: Synthetic-only imperfections limit ecological validity. How do you decide the type of the image corruption operations? Compound corruptions are the norm in practice.
>
> **A1:**
>
> **Corruption Operations & Real-World Inspiration:** Our operations are directly inspired by real-world imperfection scenarios (Examples are in Appendix A.1), and we also (already) include real-world screen-captured images (4 types, 1,028 samples).
>
> **Real-World Evaluation on Egocentric Data:** Following your suggestion, we extend V-QPP-Bench with 300 egocentric images from CRAG-MM [1] as imperfect queries, with manually cropped ground-truth entities as oracle queries, reporting Recall@3 in Table 1.
>
> **Table 1: Recall@3 on OOD Real-World Egocentric Images (CRAG-MM)**
> |OOD Type| Imperfect |V-QPP(W/O SFT)| V-QPP(SFT) |Oracle|
> |-|-|-|-|-|
> |Real-world egocentric|0.206|0.157|0.236|0.326|
>
> Results confirm: (a) imperfect images degrade retrieval; (b) oracle processing yields substantial gains; (c) our SFT model transfers effectively to real-world settings.
>
> **Multi-Step Corruption Results:** We evaluate on 1,028 images with $n$ corruption steps (1–5, random types), reporting average Recall@3 in Table 2.
>
> **Table 2: Recall@3 under Multi-Step ($n$) Corruption**
> |Step($n$)| Imperfect |Oracle| V-QPP(W/O SFT) |V-QPP(SFT)|
> |-|-|-|-|-|
> |1|0.323|0.384|0.310 |0.368|
> |3|0.150|0.284|0.152 |0.200|
> |5|0.056|0.180|0.062 |0.097|
>
> ($n$=2,4 will be included in revision.) Performance degrades as $n$ increases, confirming greater challenges under multi-step corruption. Notably, our SFT model — trained only on single-step corruptions — still meaningfully improves retrieval, demonstrating strong generalization to unseen compositional corruptions.
>
> ---
>
> > Q2: Lacking OOD analysis for SFT (unseen corruption types/real-world images).
>
> **A2:** We conduct a comprehensive OOD analysis across two challenging unseen settings:
>
> **Unseen Real-World OOD:** As shown in Table 1, V-QPP(SFT) transfers effectively to real-world egocentric images, recovering substantial performance lost to real-world imperfections.
>
> **Unseen Synthetic Configurations:** Table 3 reports results on unseen configurations: random rotation (0–360°), varied watermarks, and photo-album-style expand layouts (1,028 samples).
>
> **Table 3: Final QA Accuracy on OOD Rotation, Watermark and Expand Images**
> |OOD Type| Imperfect |V-QPP(W/O SFT)| V-QPP(SFT) |Oracle|
> |-|-|-|-|-|
> |Rotation (0–360°)|0.304|0.285|0.320|0.415|
> |Watermark (varied types/locations)|0.178 |0.179| 0.254 |0.371|
> |Expand (varied layouts/image numbers)| 0.234 |0.231| 0.437 |0.460|
>
> These results demonstrate that SFT equips the model with generalizable visual query preprocessing ability beyond simply reversing the synthetic pipeline.
>
> ---
>
> > Q3: It would strengthen the paper to show that V-QPP is complementary to or better than making the retriever itself more robust.
>
> **A3:** V-QPP operates on the raw query *before* retrieval — pixel-level manipulation analogous to word-level rewriting in text-based RAG [2][3], with the retriever treated as fixed. As shown in Figure 18 (Appendix D.2), oracle processing improves retrieval across different retrievers, confirming orthogonality to retriever optimization. The two are complementary — retrieval signals could serve as rewards for V-QPP agent training (Table 5), a promising future direction.
>
> ---
>
> > Q4: Google Lens/RL Results.
>
> **Google Lens Results:** Table 4 reports average results on 100 samples, consistent with our main findings. Per-type breakdowns will be included in the revision.  Our benchmark also readily supports larger-scale API evaluation for users with sufficient API budget.
>
> **Table 4: End-to-End QA Accuracy (%) with Google Lens on 100 InfoSeek Samples (Average)**
> |Imperfect| Oracle|V-QPP(W/O SFT)| V-QPP(SFT) |
> |-|-|-|-|
> |40.8| 56.2|35.0|53.0|
>
> **RL Results:** Our primary contribution is the benchmark and SFT baseline; advanced training methods are beyond the current scope. Nevertheless, we report average preliminary GRPO results (retrieval performance as reward, 0.5 epochs) in Table 5, leaving reward design, multi-round rollout, and alternative RL algorithms for future work.
>
> **Table 5: Preliminary RL Results (Recall@3, Average)**
> |Imperfect| Oracle |V-QPP(W/O SFT)| V-QPP(SFT) |GRPO|
> |-|-|-|-|-|
> |0.315| 0.412|0.309| 0.376 |0.382|
>
> ---
>
> > Q5: Is the SFT training set too small to learn effective preprocessing?
>
> **A5:** Our benchmark supports flexible train/test splits. Even a small training set (100 images, 1K trajectories) yields meaningful V-QPP improvements, validating task learnability.
>
> ---
>
> [1] CRAG-MM: Multi-modal Multi-turn Comprehensive RAG Benchmark.
>
> [2] Deepretrieval: Hacking real search engines and retrievers with large language models via reinforcement learning.
>
> [3] s3: You don't need that much data to train a search agent via rl.

---

> > ### Author Rebuttal · Reviewer_BFg7 · 2026-04-07
> >
> > Thanks for the rebuttal. It addressed my concerns.

---

> > > ### Author Response · Authors · 2026-04-07
> > >
> > > Dear Reviewer BFg7,
> > >
> > > Thank you for your acknowledgement. We are glad to hear that your concerns are fully resolved. Please let us know if you have any further questions before the discussion ends, and we would be happy to provide more information.
> > >
> > > We will ensure all promised revisions are included in the final paper.
> > >
> > > Best regards,
> > >
> > > Authors

---

### Decision · Program_Chairs · 2026-04-30

**Decision:**

Accept (regular)

**Comment:**

This paper introduces V-QPP-Bench, a benchmark for evaluating visual query pre-processing in Multimodal RAG, a stage analogous to query rewriting in text RAG but largely neglected in the visual domain. The rebuttal was quite successful. After rebuttal, the paper received all positive scores of 4444. Reviewers have commented that the construction of the new benchmark is well-motivated and a timely contribution. The design of the benchmark is comprehensive with extensive evaluation. Therefore, the AC would like to recommend acceptance.